# Publicly-Detectable Watermarking for Language Models

## Abstract

We present a highly detectable, trustless watermarking scheme for LLMs: the detection algorithm contains no secret information, and it is executable by anyone. We embed a publicly-verifiable cryptographic signature into LLM output using rejection sampling. We prove that our scheme is cryptographically correct, sound, and distortion-free. We make novel uses of error-correction techniques to overcome periods of low entropy, a barrier for all prior watermarking schemes. We implement our scheme and make empirical measurements over open models in the 2.7B to 70B parameter range. Our experiments suggest that our formal claims are met in practice.

## 1 Introduction

Generative AI (GenAI) technologies, such as large language models (LLMs) and diffusion models, have impressive capabilities. These capabilities include in-context learning, code completion, text-to-image generation, and document and code chat. However, GenAI technologies are also being used for nefarious purposes (e.g., generating fake tweets, generating attacks, and harmful prose). To protect against such use cases, a large body of work has focused on detecting AI-generated content (Lavergne et al., 2008; Beresneva, 2016; Gehrmann et al., 2019; Zellers et al., 2019; Mitchell et al., 2023; GPTZero, 2023; Hendrik Kirchner et al., 2023). The problem is: given content $c$, is $c$ generated by some specific GenAI technique, e.g., GPT-4 (OpenAI, 2023), Gemini (Google DeepMind, 2024), or Stable Diffusion (Rombach et al., 2022)? Informally, we want a "GenAI Turing Test."

At present, the main approach when trying to detect AI-generated text is to train yet another AI model to perform the detection (Zellers et al., 2019; Mitchell et al., 2023; GPTZero, 2023; Hendrik Kirchner et al., 2023). This method makes a critical assumption: that AI-generated text has embedded features that are identifiable by AI. The key problem with this assumption is that generative models are explicitly designed to produce realistic content that is difficult to distinguish from natural content (generated by a human or nature). As a result, any "black-box" detection scheme will suffer from high false positive and/or false negative rates as generative models improve. Available detectors such as GPTZero (GPTZero, 2023) have no guarantee of correctness—e.g., the authors state outright that detection results from their tool should not be used to punish students.

To circumvent this fundamental issue, a recent line of work (Aaronson, 2023; Kirchenbauer et al., 2023; Christ et al., 2023; Kuditipudi et al., 2023) has taken a different approach to detecting AI content. These watermarking techniques alter the generation process to embed a "signal" in the generated content. The detection process measures the signal: if the signal is sufficiently strong, the content was likely watermarked. In particular, the cryptographic approach of Christ et al. (2023) achieves formal notions of completeness (any watermarked text will be detected), soundness (one cannot watermark a text without knowing the secret), and distortion-freeness (watermarking does not change the output distribution). Finally, these watermarking schemes are "keyed" in the sense that the signal is a function of a secret key. The same key is used to generate and measure the signal.

The aforementioned watermarking approaches have one problem in common: the model provider and the detector both need to know a shared secret key. This is acceptable in scenarios where the entity trying to detect the watermark is the same entity generating the content. For example, an entity that provides a chat API may be able to provide a detection API as well. However, such a setup has limitations:

1. **Lack of privacy:** The entity who wants to check the integrity of the content might not be willing to share it with the detector. For example, one looking to identify whether their medical records are AI-generated may not want to share the records themselves.

2. **Conflict of interest:** The entity providing the detection API might not be trusted in certain cases. For instance, consider a case where the entity is accused of generating a certain inappropriate text and is brought to a court of law. It is not reasonable to ask the same entity to tell whether the text is watermarked.

One solution could be sharing the secret with the world so everyone can run the detection. However, this raises another important problem: anyone can now embed the secret to any content, AI-generated or not. This would not be acceptable because the watermarking is subject to denial of service attacks. An attacker can create masses of watermarked content that is not AI-generated to undermine the dependability of the detector. Consider the effect on one of the main applications of watermarking: an entity may want to use the watermark as a signature for their content. Such signatures are useful when (a) the generated content needs to come with proof of a credible generator, and (b) the entity needs to refute an accusation about a generated content; i.e., it should not be accountable for a content without its watermark. This application is rendered impossible in a world with availability attacks.

In this paper, we aim to solve the aforementioned problems for LLMs that produce text. We ask:

> Is it possible to construct a *publicly-detectable* watermarking scheme with cryptographic detectability and distortion-freeness?

We find that the answer is yes: we construct a publicly-detectable scheme that provably resolves the trust issue—users can cryptographically verify the presence of a watermark. Further, they have a guarantee that the only entity capable of embedding the watermark is the model provider, resolving the privacy and conflict of interest issues above. We state the properties for public detectability below:

1. **Cryptographic detectability:** To guarantee a user is convinced that a watermark is detected, the watermarking scheme must achieve cryptographic detectability: false positives or negatives must never occur in practice.

2. **Weak robustness:** It is possible that text obtained from LMs is modified—to some extent—before publication. The watermark detector should be able to detect a watermark so long as the cryptographic signature is still embedded in the text. Prior work in the secret key setting aimed for *strong* robustness where detection should be possible even if the LLM output has changed substantially but text semantics are preserved. Strong robustness has since been shown to be impossible in the general case (Zhang et al., 2023) and we focus on ensuring high detectability as a first step.

3. **Distortion-freeness:** The watermarking scheme should not degrade the quality of the LLM output. No probabilistic polynomial-time (PPT) adversary should be able to distinguish between watermarked and non-watermarked text.

4. **Model agnosticity:** The watermarking scheme should use the model as a black box, i.e., it should not rely on any specific model weights or configurations.

5. **Public verifiablity:** Without access to the model weights or secret material of the watermarking scheme, the detector should still be able to determine whether a candidate text is watermarked.

## 2 Technical Overview

We give an overview of the key ideas in our construction. Refer to Figure 1 for a visual representation and Section 4 for full details.

Let $\mathbf{t} := t_1, t_2, \ldots, t_\ell$ be bit samples from probability distributions $p_1, p_2, \ldots, p_\ell$ where each $p_i$ is a probability distribution from an auto-regressive model. Our scheme assumes that any consecutive $\ell$ tokens output by

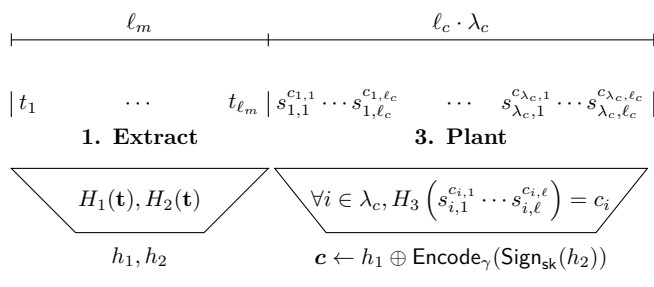

Figure 1: Our core gadget. Embedding is a three-step process as designated by (1) through (3). First (1), $\ell_m$ tokens are sampled natively from the LLM. These tokens $\boldsymbol{t}$ are hashed twice with two different hash functions, producing $h_1 \leftarrow H_1(\boldsymbol{t})$ and $h_2 \leftarrow H_2(\boldsymbol{t})$. Second (2), $h_2$ is signed with the secret key sk, error-corrected, and randomized with $h_1$. The final product is a pseudorandom bitstring $\boldsymbol{c} \leftarrow h_1 \oplus \mathsf{Encode}_\gamma(\mathsf{Sign}_{\mathsf{sk}}(h_2))$. Lastly (3), each bit $c_i$ the randomized codeword is embedded into the next $\ell_c$ tokens by rejection sampling. That is, the $i$-th block of $\ell_c$ tokens are sampled such that the hash of the block yields the $i$-th bit of the randomized codeword, i.e., $\forall i \in \lambda_c, H_3\left(s_{i,1}^{c_{i,1}} \cdots s_{i,\ell_c}^{c_{i,\ell_c}}\right) = c_i$ where each $s$ is one token.

the language model contain sufficient entropy (formally captured by Assumption 3.1). Hence, we know that $\sum_{i=1}^{\ell} -\ln p_i(t_i) \geq \alpha$ for some reasonably large $\alpha$. That is, the $\ell$ tokens were sampled from distributions with at least $\alpha$ cumulative bits of entropy. Let $\mathbf{t}$ denote the first $\ell$ tokens sampled from the model (denote $\mathbf{t}$ as the message) and let $\boldsymbol{\sigma} := \mathsf{Sign}_{\mathsf{sk}}(H(\mathbf{t}))$.[1] We can embed the $\lambda_\sigma$-bit signature $\boldsymbol{\sigma} = \sigma_1, \sigma_2, \ldots, \sigma_{\lambda_\sigma}$ in a contiguous sequence of tokens from the auto-regressive model as follows: for each of the next $\ell \cdot \lambda_\sigma$ tokens sampled from the model, ensure that the $i$-th block of $\ell$ tokens hashes to the corresponding $i$-th bit in $\boldsymbol{\sigma}$, i.e., $H(t_{i+1}, t_{i+2}, \ldots, t_{i+\ell}) = \sigma_i$ for $i \in [\lambda_\sigma]$. After this process, a complete message-signature pair is embedded into a contiguous sequence of generated tokens. We remark that our watermarked output is (computationally) indistinguishable from the original output: as long as there is sufficient entropy at generation time, no PPT algorithm can tell if a text completion came from the watermarking algorithm or the plain algorithm.

To detect the presence of a watermark, the detector needs to recover the message-signature pair. The detector first recovers the message $\mathbf{t}$ by looking at the first $\ell$ tokens. Next, the detector recovers each bit of the signature by computing $\sigma_i = H(t_{i+1}, t_{i+2}, \ldots, t_{i+\ell})$ for $i \in [\lambda_\sigma]$ and let $\boldsymbol{\sigma} = (\sigma_1, \ldots, \sigma_{\lambda_\sigma})$. It can then verify the signature by computing $\mathsf{Verify}_{\mathsf{pk}}(H(\mathbf{t}), \boldsymbol{\sigma})$ using the public verification key: if the signature verifies, then the text was watermarked.

## 2.1 Dealing with low entropy sequences

As in the private key setting, our protocol needs to handle sequences with limited entropy. Kaptchuk et al. (2021) provide an illustrative example: given the inputs "The largest carnivore of the Cretaceous period was the Tyrannosaurus," the next token is almost certainly going to be "Rex." Assuming that "Rex" is a whole token and it does not hash to the desired bit, text generation cannot continue.

We overcome this problem by leveraging standard error correction. Instead of embedding $\boldsymbol{\sigma} := \mathsf{Sign}_{\mathsf{sk}}(H(\mathbf{t}))$ directly, we can instead embed $\mathbf{c} := \mathsf{Encode}_\gamma(\boldsymbol{\sigma})$ where $\mathbf{c}$ is a codeword of length $\lambda_c > \lambda_\sigma$ that allows for correction of up to $\gamma$ errors. Now, at generation time, we can tolerate up to $\gamma$ periods of low entropy—when such a scenario is encountered, we can plant tokens that do not satisfy the rejection sampling condition. At detection time, we can correct these planted errors so long as they do not exceed the maximum amount $\gamma$.

---

[1]Here, $H$ is a cryptographic hash function and $\mathsf{Sign}$ is the signing algorithm of a digital signature scheme (refer to Section 4).

# 3 Security Model

This section defines what it means for a publicly-detectable watermarking scheme to be secure. We will eventually prove that our construction satisfies these definitions.

## 3.1 Preliminaries

Let $a \parallel b$ denote the concatenation of $a$ to $b$. We use $\log(\cdot)$ to take logarithms base 2. Let $\epsilon$ denote an empty list or empty string. Let $a_i$ denote the $i$-th bit of vector $\mathbf{a}$. We use Python slicing notation throughout: $\mathbf{a}[-i]$ refers to the $i$-th last element of a list and $\mathbf{a}[j:k]$ extracts the elements $a_i$ for $i \in [j, k)$. We use $\mathcal{U}$ to represent the uniform distribution and $\overset{\$}{\leftarrow}$ to denote a random sample, e.g., $r \overset{\$}{\leftarrow} \mathcal{U}$.

For the cryptographic primitives in this paper, we use $\lambda$ for the security parameter. A negligible function $\mathsf{negl}(\lambda)$ in $\lambda$ are those functions that decay faster than the inverse of any polynomials. That is, for all $\mathsf{poly}(\lambda)$, it holds that $\mathsf{negl}(\lambda) < \frac{1}{\mathsf{poly}(\lambda)}$ for all large enough $\lambda$.

## 3.2 Assumptions

We assume that any contiguous block of $\ell$ tokens contains at least $\alpha$ bits of min-entropy, i.e., no particular sample is more than $2^{-\alpha}$ likely to happen.[2] This assumption allows us to capture security properties and present our protocol concisely. In addition, $\ell$ effectively serves as a parameter to tune the trade-off between robustness and distortion-freeness. Higher $\ell$ values lead to more distortion-free text at the cost of robustness and vice versa.

**Assumption 3.1.** For any prompt $\boldsymbol{\rho}$ and tokens $\boldsymbol{t}$, the new tokens $\boldsymbol{t}' \leftarrow \mathsf{GenModel}_\ell(\boldsymbol{\rho}, \boldsymbol{t}) \in \mathcal{T}^\ell$ were sampled from distributions with min-entropy at least $\alpha$.

If this assumption is met, distortion-freeness is guaranteed. However, our construction makes novel use of error-correcting codes (ECC) to weaken the entropy requirement in practice—our protocol can tolerate a fixed number of periods where the min entropy is *below* $\alpha$. The maximum number of low-entropy periods our scheme can tolerate is exactly the maximum number of errors that the underlying ECC scheme can correct.

## 3.3 Entity Interaction

We refer to two distinct entities in our security model:

**Model provider** The model provider provides the LLM service: given a prompt, it returns the LLM output for that prompt and the given LLM configuration. An honest model provider will run the watermarking protocol at text generation time. This entity has white-box access to the model weights in addition to any secret material specific to the watermarking protocol, e.g., a secret watermarking key.

**User** Users generate prompts which are sent to the model provider in exchange for the model output. Users may test text for the presence of a watermark by running the detection algorithm on candidate text and an LLM provider's public key. The user should be convinced that the watermark is present or not, i.e., the detector must provide a "proof of watermark" that can be verified without model weights or secret material pertaining to the watermarking protocol.

## 3.4 Definitions

In this section, we formally define a publicly detectable watermarking scheme, which should satisfy (a) completeness, (b) soundness, (c) distortion-freeness, and (d) robustness. We prove our scheme meets these definitions.

---

[2]Formally, the min-entropy $H_\infty(\mathcal{D})$ of a distribution $D$ is defined as $-\log\left(\max_{\omega \in \mathsf{Supp}(\mathcal{D})} \Pr[D = \omega]\right)$.

**Definition 3.2** (Publicly-Detectable Watermarking Scheme)**.** A publicly-detectable watermarking scheme PDWS for an auto-regressive model Model over token vocabulary $\mathcal{T}$ is a tuple of algorithms PDWS = (Setup, Watermark, Detect) where:

- Setup$(1^\lambda) \to (\mathsf{sk}, \mathsf{pk})$ outputs a public key pair $(\mathsf{sk}, \mathsf{pk})$ with respect to the security parameter $\lambda$.

- Watermark$_{\mathsf{sk}}(\boldsymbol{\rho}) \overset{\$}{\to} \boldsymbol{t}$ produces response text $\boldsymbol{t} \in \mathcal{T}^*$ given a prompt $\boldsymbol{\rho} \in \mathcal{T}^*$ using the secret key $\mathsf{sk}$.

- Detect$_{\mathsf{pk}}(\boldsymbol{t}^*) \to \{\texttt{true}, \texttt{false}\}$ outputs $\texttt{true}$ or $\texttt{false}$ given a candidate watermarked text $\boldsymbol{t}^*$.

A PDWS scheme is considered secure if the following security definitions are met.

**Definition 3.3** (Completeness)**.** A PDWS is $\delta$-complete if for every prompt $\boldsymbol{\rho}$ and token sequence $\boldsymbol{t} \in \mathcal{T}^*$ of length $|\boldsymbol{t}| \geq \delta$, it holds that

$$\Pr_{\substack{(\mathsf{sk},\,\mathsf{pk})\,\leftarrow\,\mathsf{Setup}(1^\lambda) \\ \boldsymbol{t}\,\leftarrow\,\mathsf{Watermark}_{\mathsf{sk}}(\boldsymbol{\rho})}} \left[\mathsf{Detect}_{\mathsf{pk}}(\boldsymbol{t}) = \texttt{false}\right] \leq \mathsf{negl}(\lambda).$$

The $\delta$-completeness ensures that text of sufficient length that was watermarked with the honest protocol results in non-detection with negligible probability. This definition is an asymmetric-key analogue of the symmetric-key completeness definition in Christ et al. (2023).

**Definition 3.4** (Soundness/Unforgeability)**.** A PDWS is $k$-sound if any adversary $\mathcal{A}$ cannot generate a watermarked text given the public detection key and any polynomial number of genuinely-watermarked texts. Let $E$ be the event that

$$\left\{ \begin{array}{c} \mathsf{Detect}_{\mathsf{pk}}(\boldsymbol{t}^*) = \texttt{true} \,\wedge \\ \mathsf{non\_overlapping}_k(\boldsymbol{t}^*, \boldsymbol{t}_1, \boldsymbol{t}_2, \ldots) = \texttt{true} \end{array} \right\}$$

where the predicate $\mathsf{non\_overlapping}_k(\boldsymbol{t}^*, \boldsymbol{t}_1, \boldsymbol{t}_2, \ldots)$ outputs $\texttt{true}$ if $\boldsymbol{t}^*$ does not share a $k$-length window of tokens with any of the genuinely-watermarked texts $\boldsymbol{t}_1, \boldsymbol{t}_2, \ldots$ and outputs $\texttt{false}$ otherwise. We define soundness as

$$\Pr_{\substack{(\mathsf{sk},\,\mathsf{pk})\,\leftarrow\,\mathsf{Setup}(1^\lambda) \\ \boldsymbol{t}^*\,\leftarrow\,\mathcal{A}^{\mathsf{Watermark}_{\mathsf{sk}}(\cdot)}(\mathsf{pk})}} \left[E\right] \leq \mathsf{negl}(\lambda).$$

Here, the adversary is allowed to make a polynomial number of queries to the oracle Watermark$_{\mathsf{sk}}(\cdot)$, and we use $\boldsymbol{t}_1, \boldsymbol{t}_2, \ldots$ to denote the watermarked text that the adversary receives as output when she queries the model Watermark$_{\mathsf{sk}}(\cdot)$.

**On the unforgeability of our scheme** Intuitively, our soundness definition says the following. If the adversary manages to output a text $\boldsymbol{t}^*$ that is labeled as watermarked, it must be the case that she copied a sufficiently long sequence of tokens from the genuinely-watermarked texts she received from the model (i.e., $\boldsymbol{t}_1, \boldsymbol{t}_2, \ldots$). This implies that any attempted forgery of a watermarked message must contain an overwhelming portion of tokens from genuine watermarked text. We emphasize that this notion of unforgeability is *parametrized* (by the overlapping length $k$). Intuitively, the larger $k$ is the more sound our scheme is. Looking ahead, our main construction is flexible in that, for any desired overlapping parameter $k$, our construction can be adapted to meet the corresponding soundness guarantee.

**Definition 3.5** (Auto-regressive Model)**.** An auto-regressive model Model over token vocabulary $\mathcal{T}$ is a deterministic algorithm that takes in a prompt $\boldsymbol{\rho} \in \mathcal{T}^*$ and tokens previously output by the model $\boldsymbol{t} \in \mathcal{T}^*$ and outputs a probability distribution $p = \mathsf{Model}(\boldsymbol{\rho}, \boldsymbol{t})$ over $\mathcal{T}$.

GenModel wraps around Model to implement a generative model as shown in Algorithm 1. We use Model and GenModel for subsequent definitions and proofs.

---

**Algorithm 1** GenModel

---

1: **input:** $n$, $\boldsymbol{\rho}$,
2: $\boldsymbol{t} \leftarrow \epsilon$
3: **for** $i = 1$ **to** $n$ **do**
4: $\quad \boldsymbol{t} \leftarrow \boldsymbol{t} \parallel \mathsf{Decode}(\mathsf{Model}(\boldsymbol{\rho}, \boldsymbol{t}))$
5: **end for**
6: **output:** $\boldsymbol{t}$

---

GenModel iteratively generates $n$ tokens. Decode is the specific decoding method. Throughout this paper, we fix Decode to multinomial sampling, though any decoding algorithm that satisfies Assumption 3.1 would suffice.

**Definition 3.6** (Distortion-freeness). A PDWS is (computationally) $\epsilon$-distortion-free if, for all PPT distinguishers $D$,

$$\left| \Pr\left[ D^{\mathsf{Model},\mathsf{GenModel}}(1^\lambda) \to 1 \right] - \Pr_{(\mathsf{sk},\mathsf{pk}) \leftarrow \mathsf{Setup}(1^\lambda)}\left[ D^{\mathsf{Model},\mathsf{Watermark_{sk}}}(1^\lambda) \to 1 \right] \right| \leq \epsilon.$$

This means distortion-freeness ensures that the watermarking algorithm does not noticeably change the quality of the model output, i.e., without the secret watermarking key, no PPT machine can distinguish plain LLM output from watermarked LLM output. This definition is the same as Christ et al. (2023)'s undetectability definition—we denote it as distortion-freeness to better reflect the property it captures in our setting.

**Definition 3.7** (Robustness). A publicly-detectable watermarking scheme is $\delta$-robust if, for every prompt $\boldsymbol{\rho}$ and security parameter $\lambda$,

$$\Pr_{\substack{(\mathsf{sk},\, \mathsf{pk}) \leftarrow \mathsf{Setup}(1^\lambda) \\ \boldsymbol{t} \leftarrow \mathsf{Watermark_{sk}}(\boldsymbol{\rho})}} \left[ \mathsf{Detect_{pk}}(\mathcal{A}(\boldsymbol{t})) = \texttt{false} \right] \leq \mathsf{negl}(\lambda)$$

where the adversary is allowed to transform the input text $\boldsymbol{t}$ however she pleases so long as a $\delta$-length contiguous sequence of tokens remains. Formally, let $\boldsymbol{t}^*$ be the adversarially-modified text (i.e. $\boldsymbol{t}^* \leftarrow \mathcal{A}(\boldsymbol{t})$). Then, there must exist a $\delta$-length window of tokens in $\boldsymbol{t}^*$ that exactly matches a $\delta$-length window in $\boldsymbol{t}$.

## 4 Protocol

Our scheme uses a public-key signature scheme with the following properties.

**Definition 4.1** (Public-Key Signature Scheme). A public-key signature scheme $\mathsf{S}$ is a tuple of algorithms $\mathsf{S} = (\mathsf{Gen}, \mathsf{Sign}, \mathsf{Verify})$ where:

- $\mathsf{Gen}(1^\lambda) \to (\mathsf{sk}, \mathsf{pk})$ outputs a key pair $(\mathsf{sk}, \mathsf{pk})$ with respect to the security parameter $\lambda$.

- $\mathsf{Sign_{sk}}(m) \to \sigma$ produces a signature $\sigma$, given a message $m$, using the secret signing key $\mathsf{sk}$.

- $\mathsf{Verify_{pk}}(m, \sigma) \to \{\texttt{true}, \texttt{false}\}$ outputs $\texttt{true}$ or $\texttt{false}$, given a candidate message $m$ and signature $\sigma$, using the public verification key.

**Definition 4.2** (Unforgeability). For every adversary $\mathcal{A}$, we have

$$\Pr_{\substack{(\mathsf{pk},\, \mathsf{sk}) \leftarrow \mathsf{Gen}(1^\lambda) \\ (m^*, \sigma^*) \leftarrow \mathcal{A}^{\mathsf{Sign_{sk}}(\cdot)}(\mathsf{pk})}} \left[ \mathsf{Verify_{pk}}(m^*, \sigma^*) = \texttt{true} \right] \leq \mathsf{negl}(\lambda).$$

Here, the adversary gets oracle access to the signing oracle $\mathsf{Sign_{sk}}(\cdot)$, but $m^*$ in the final forgery output $(m^*, \sigma^*)$ must have never been queried using the signing oracle. As a signature scheme, we require this property to guarantee it is hard to forge a watermark.

**Definition 4.3** (Hamming Distance). For alphabet $\Sigma$ and $x, y \in \Sigma^n$, define the Hamming distance between $x$ and $y$ as

$$\mathsf{Hamming}(x, y) := |\{i \in [n] : x_i \neq y_i\}|.$$

**Definition 4.4** (Error-Correcting Code). For an alphabet $\Sigma$, an $[n, k, d]_\Sigma$ error-correcting code is a 2-tuple $(\mathsf{Encode}, \mathsf{Decode})$ algorithm where $\mathsf{Encode} : \Sigma^k \to \Sigma^n$ is an encoding algorithm such that for all $m, m' \in \Sigma^k$ where $m \neq m'$,

$$\mathsf{Hamming}(\mathsf{Encode}(m), \mathsf{Encode}(m')) \geq d$$

and $\mathsf{Decode} : \Sigma^n \to \Sigma^k$ is the decoding algorithm such that, for all message $m \in \Sigma^k$ and *erroneous* codeword $c \in \Sigma^n$, we have

$$\mathsf{Hamming}(\mathsf{Encode}(m), c) \leq \gamma_{\max} \implies \mathsf{Decode}(c) = m$$

where $\gamma_{\max} \leq (d - 1)/2$ is the maximum number of erroneous symbols that $\mathsf{Decode}$ can correct.

## 4.1 Private Generation Algorithm

We present our high-level watermarking scheme in Algorithm 2. The core idea is to embed a message and a corresponding publicly-verifiable signature in the generated text. The message-signature pair should be extractable during detection. Once extracted, it can be verified using the public key.

To explain our scheme, we describe how to embed one message-signature pair in LLM output—the construction can be applied repeatedly to generate arbitrarily long LLM output (i.e., Line 4 in Algorithm 2). Refer to Figure 1 for a simplified visual presentation of the construction.

The first step is to sample a fixed number of tokens such that the entropy used at generation time to produce those tokens is sufficient for watermarking. This is captured in Line 2 of Algorithm 3. By Assumption 3.1, we know that $\ell$ tokens were sampled from distributions with at least $\alpha$ bits of entropy. Denote these $\ell$ tokens as the message $\boldsymbol{t}$. Once $\boldsymbol{t}$ has been sampled, it is hashed, signed, and error-corrected (Lines 3-4). Now, any error-correcting codeword is not a pseudorandom string; therefore, directly embedding a codeword distorts the distribution of the output. However, we can regain pseudorandomness by using the message hash as a one-time pad to mask the codeword. Specifically, we encode $\mathbf{c} := H_2(m) \oplus \mathsf{Encode}(\sigma)$ where $H_2(\cdot)$ is a different hash function than the one used to originally hash the message since $H_1(\cdot)$ and $H_2(\cdot)$ map to a different range of bits.

Once the pseudorandom signature codeword $\mathbf{c}$ is computed, the next step is to embed it into natural language. The key idea is to embed bits into a block of tokens such that the block of tokens hashes to the target bit. In particular, the construction embeds $\beta$ bits into each $\ell$ tokens. In Lines 4-16 in Algorithm 4, we sample $\ell$ more tokens using the native LLM decoder and check if there is a hash collision. In particular, we try $a_{\max}$ times to find the best next $\ell$ tokens that hash to the next $\beta$ embedded bits, where optimality is measured by Hamming distance. Note that the hash depends on all previous inputs to hashes for the current signature codeword. Once we find the optimal output, we accept the token block and move on to the next $\beta$ bits of the signature codeword. Otherwise, reject the tokens and freshly sample a new block of length $\ell$. At the end of the rejection sampling process, the signature will be embedded in $\ell \cdot \lambda_c$ tokens where $\lambda_c$ is the length of the signature codeword—one message-signature pair is embedded in generated text. This process can be repeated to embed multiple pairs for added resilience.

## 4.2 Public Detection Algorithm

To detect if a watermark is present in candidate text, it suffices to extract one message-signature pair and verify it using the public key. In Line 2 in Algorithm 5, we iterate over all potential token blocks of length $\ell$ (adjusting by $\lambda_c = \frac{\lambda_\sigma}{\beta}$ to account for the signature codeword length). Once the message $\mathbf{m}$ is assigned, the signature is iteratively reconstructed in Lines 5-9. Notably, since we employ an error-correcting code to handle the cases where the entropy is low to embed bits, we must invoke the error-correction algorithm

---

**Algorithm 2** Watermark

1: **constants:** sk, $n$, $\ell$, $\lambda_c$, $\beta$, $a_{\max}$, $\gamma_{\max}$
2: **input:** $\boldsymbol{\rho}$
3: $\boldsymbol{t} \leftarrow \epsilon$
4: **while** $|\boldsymbol{t}| + \ell \cdot \lambda_c < n$ **do**
5: $\quad \boldsymbol{t} \leftarrow \mathsf{GenerateMessageSignaturePair}(\boldsymbol{t})$
6: **end while**
7: **if** $|\boldsymbol{t}| < n$ **then**
8: $\quad \boldsymbol{t} \leftarrow \boldsymbol{t} \parallel \mathsf{GenModel}_{n-|\boldsymbol{t}|}(\boldsymbol{\rho}, \boldsymbol{t})$
9: **end if**
10: **output:** $\boldsymbol{t}$

---

Watermark is the main watermarking algorithm. It generates a text completion for input prompt $\boldsymbol{\rho}$ consisting of $n$ watermarked tokens.

---

**Algorithm 3** GenerateMessageSignaturePair

1: **input:** $\boldsymbol{t}$
2: $\boldsymbol{t} \leftarrow \boldsymbol{t} \parallel \mathsf{GenModel}_\ell(\boldsymbol{\rho}, \boldsymbol{t})$
3: $\boldsymbol{\sigma} \leftarrow \mathsf{Sign}_{\mathsf{sk}}(H_1(\boldsymbol{t}[-\ell :]))$
4: $\mathbf{c} \leftarrow H_2(\boldsymbol{t}[-\ell :]) \oplus \mathsf{Encode}_\gamma(\boldsymbol{\sigma})$
5: $\mathbf{m}, \mathbf{c}_{\mathsf{prev}} \leftarrow \epsilon, \epsilon$
6: $\gamma \leftarrow 0$
7: **while** $\mathbf{c} \neq \epsilon$ **do**
8: $\quad c, \mathbf{c} \leftarrow \mathbf{c}[0 : \beta], \mathbf{c}[\beta :]$
9: $\quad \boldsymbol{t}, \mathbf{m}, \mathbf{c}_{\mathsf{prev}} \leftarrow \mathsf{RejectSampleTokens}(c, \boldsymbol{t}, \mathbf{m}, \mathbf{c}_{\mathsf{prev}})$
10: **end while**
11: **output:** $\boldsymbol{t}$

---

GenerateMessageSignaturePair plants the message-signature pair gadget into $\ell \cdot \lambda_c$ tokens. First, the message is sampled naively from the underlying model and the error-corrected signature $\mathbf{c}$ is computed. $\mathbf{c}$ is then iteratively embedded into $\ell \cdot \lambda_c$ tokens using rejection sampling.

---

**Algorithm 4** RejectSampleTokens

1: **input:** $c, \boldsymbol{t}, \mathbf{m}, \mathbf{c}_{\mathsf{prev}}$
2: $a \leftarrow 0$
3: $\mathbf{x}_{\mathsf{best}}, d_{\mathsf{best}} \leftarrow \epsilon, \infty$
4: **repeat**
5: $\quad \mathbf{x} \leftarrow \mathsf{GenModel}_\ell(\boldsymbol{\rho}, \boldsymbol{t})$
6: $\quad a \leftarrow a + 1$
7: $\quad d \leftarrow \mathsf{Hamming}(H_1(\mathbf{m} \parallel \mathbf{x} \parallel \mathbf{c}_{\mathsf{prev}}), c)$
8: $\quad$ **if** $d < d_{\mathsf{best}}$ **then**
9: $\quad\quad d_{\mathsf{best}}, \mathbf{x}_{\mathsf{best}} \leftarrow d, \mathbf{x}$
10: $\quad$ **end if**
11: $\quad$ **if** $(a > a_{\max} \,\wedge\, \gamma < \gamma_{\max})$ **then**
12: $\quad\quad \mathbf{x} \leftarrow \mathbf{x}_{\mathsf{best}}$
13: $\quad\quad \gamma \leftarrow \gamma + 1$
14: $\quad\quad$ **break**
15: $\quad$ **end if**
16: **until** $H_1(\mathbf{m} \parallel \mathbf{x} \parallel \mathbf{c}_{\mathsf{prev}}) = c$
17: $\mathbf{m} \leftarrow \mathbf{m} \parallel \mathbf{x}$
18: $\boldsymbol{t} \leftarrow \boldsymbol{t} \parallel \mathbf{x}$
19: $\mathbf{c}_{\mathsf{prev}} \leftarrow \mathbf{c}_{\mathsf{prev}} \parallel c$
20: **output:** $\boldsymbol{t}, \mathbf{m}, \mathbf{c}_{\mathsf{prev}}$

---

RejectSampleTokens controls the rejection sampling loop. It generates $\ell$ tokens such that each contiguous block of $\ell$ tokens encodes $c$: one bit of information.

---

**Algorithm 5** Detect

---

1: **input:** pk, $n$, $\ell$, $\lambda_c$, $\beta$, $\gamma$, $\boldsymbol{t}'$
2: **for** $i \in [n - (\ell + \lambda_c)]$ **do**
3: $\quad \boldsymbol{t} \leftarrow H_1(\boldsymbol{t}'[i : i + \ell])$
4: $\quad \mathbf{m}, \mathbf{c} \leftarrow \epsilon, \epsilon$
5: $\quad$ **for** $j \in [\lambda_c]$ **do**
6: $\quad\quad \mathbf{m} \leftarrow \mathbf{m} \parallel \boldsymbol{t}'[(i + \ell) + j \cdot \ell + 1 : (i + \ell) + (j + 1) \cdot \ell]$
7: $\quad\quad \mathbf{c} \leftarrow \mathbf{c} \parallel H_1(\mathbf{m} \parallel \mathbf{c}))$
8: $\quad$ **end for**
9: $\quad \boldsymbol{\sigma} \leftarrow \mathsf{Decode}_\gamma(H_2(\boldsymbol{t}'[i : i + \ell]) \oplus \mathbf{c})$
10: $\quad$ **if** $\mathsf{Verify}_{\mathsf{pk}}(\mathbf{m}, \boldsymbol{\sigma}) = \texttt{true}$ **then**
11: $\quad\quad$ **output:** `true`
12: $\quad$ **end if**
13: **end for**
14: **output:** `false`

---

Detect is the watermark detection algorithm. Given potentially watermarked text $\boldsymbol{t}'$, it exhaustively searches for an embedded message-signature pair that passes authentication. If one such pair is found, the input text is flagged as watermarked.

to correctly decode the signature embedded in the (potentially) erroneous codeword. This is exactly what Line 9 does. If the signature verifies, we know with overwhelming probability the text was watermarked (See Theorem 4.6). Otherwise, move on to the next candidate block and try again. If no message-signature pair is verified, we conclude that the text was not watermarked (See Theorem 4.7)

### 4.3 Formal Guarantees of Our Construction

Our construction uses random oracle $\mathcal{O}$ to model a cryptographic hash function $H$. A random oracle is a random function drawn uniformly randomly from the set of all possible functions (over specific input and output domains). Random oracle models are commonly used in cryptographic construction (Bellare & Rogaway, 1993). Constructions that are provably secure in the random oracle model are heuristically assumed to be also secure when one instantiates the random oracle $\mathcal{O}$ with a cryptographic hash function $H$. We use $\mathcal{O}$ and $H$ interchangeably in the proof.

Assume that each block of $\ell$ tokens has at least $\alpha$ bits of entropy. We model the cryptographic hash $H(\cdot)$ as a random oracle, denoted $\mathcal{O}(\cdot)$. Without loss of generality, the proof is written for the case of $\beta = 1$, i.e., we embed one bit into each $\ell$ tokens. It generalizes to any $\beta$.

**Theorem 4.5** (Distortion-freeness). *Let $H_1, H_2$ be random oracles. $\mathcal{PDWS}$ is a computationally distortion-free publicly detectable watermarking scheme assuming every $\ell$ tokens generated by the LLM contains $\alpha$ bits of entropy.*

*Proof.* Our proof relies on the following claim.

*Claim* 1 (Balanced Partition). Let $\mathcal{D}$ be any distribution with min-entropy $\geq \alpha$. It holds that

$$\Pr_{H_1}\left[\left|\Pr_{\mathcal{D}}[H_1(\mathcal{D}) = 1] - 1/2\right| \geq \frac{1}{2} \cdot \sqrt{\alpha} \cdot 2^{-\alpha/2}\right] \leq 2 \cdot 2^{-\alpha}.$$

That is, a randomly sampled hash function $H_1$ will result in a balanced bi-partition $H_1^{-1}(0)$ and $H_1^{-1}(1)$ on the support of $\mathcal{D}$ with overwhelming probability.

*Proof of Claim 1.* Since $\mathcal{D}$ contains at least $\alpha$ bits of min-entropy, the support of $\mathcal{D}$ contains at least $2^{\alpha}$ elements; let us denote them by $x_1, \ldots, x_u$, where $u \geq 2^{\alpha}$. Let $X_i$ be the random variable defined as

$$X_i = \begin{cases} \Pr[\mathcal{D} = x_i] & \text{when } H_1(x_i) = 1 \\ 0 & \text{when } H_1(x_i) = 0 \end{cases}.$$

Clearly, $\Pr_{\mathcal{D}}[H_1(\mathcal{D}) = 1] = \sum_{i=1}^{u} X_i$. Observe that $X_i$ are independent random variables satisfying $0 \leq X_i \leq 2^{-\alpha}$. By the Hoeffding inequality (Theorem A.1), we have

$$\Pr_{H_1}\left[\left|\Pr_{\mathcal{D}}[H_1(\mathcal{D}) = 1] - 1/2\right| \geq \frac{1}{2} \cdot \sqrt{\alpha} \cdot 2^{-\alpha/2}\right] \leq 2 \cdot 2^{-\frac{\alpha \cdot 2^{-\alpha}}{2^{-\alpha}}} = 2 \cdot 2^{-\alpha}.$$

Here, we use the fact that $\sum_i (b_i - a_i)^2 = \sum_i b_i^2 \leq (\max_i b_i) \cdot \sum_i b_i = \max_i b_i \leq 2^{-\alpha}$. This completes the proof. $\square$

Now we proceed to prove our theorem. The only difference between our sampling algorithm and the original sampling algorithm is the following: The original sampling algorithm (i.e., multinomial sampling) samples the next $\ell$ tokens directly from some distribution $\mathcal{D}$. Our sampling algorithm first samples a bit $b$ and then samples the next batch of $\ell$ tokens according to $\mathcal{D}$, but conditioned on that its hash is consistent with $b$. We just need to prove that these two sampling processes are computationally indistinguishable.

By the randomness of the output of the random oracle $H_2$, each embedded bit $b$ is computationally indistinguishable from a truly random bit. Therefore, it suffices to prove that $\mathcal{D}$ is close to first sampling a truly random bit $b$ and then sampling from $\mathcal{D}$ conditioned on the hash being $b$.

Observe that, if $H_1^{-1}(0)$ and $H_1^{-1}(1)$ gives a *perfect* balanced partition on the support of $\mathcal{D}$ (i.e., the probability of the hash of a sample from $\mathcal{D}$ is perfectly uniformly random), then these two ways of sampling from $\mathcal{D}$ is *identical*. Now, our Claim 1 states that, for every $\ell$ tokens that the LLM outputs, as long as it contains $\alpha$ bits of entropy, a randomly sampled hash function $H_1$ will *not* give a well-balanced partition with *exponentially small probability.* Suppose the LLM outputs a total of $m$ sets of $\ell$ tokens. By a simple union bound over all such sets, a randomly sampled hash function $H_1$ will give a well-balanced partition on *all these $m$ distributions* with probability $1 - m \cdot \exp(-\Omega(\alpha))$. Conditioned on that hash function gives a well-balanced distribution on all these $m$ distributions, the distribution that our sampling process gives and the distribution that the original LLM outputs are indeed $\exp(-\Omega(\alpha))$-close by our Claim 1.

This completes the proof that our sampling process is computationally indistinguishable from the original sampling process; hence, our scheme is computationally $\epsilon$-distortion-free, where $\epsilon = \exp(-\Omega(\alpha))$. $\square$

**Theorem 4.6** (Completeness). *$\mathcal{PDWS}$ is a $(\ell + \ell \cdot \lambda_\sigma)$-complete publicly detectable watermarking scheme.*

*Proof.* For any long enough output $\boldsymbol{t}$, it is easy to see that if the watermarking scheme successfully embeds in a message/signature pair, the detection algorithm will mark the text as "watermarked". The only possibility that the watermarking fails is if the rejection sampling algorithm fails to find the next batch of tokens whose hash is consistent with the target bit.

For any $\ell$ consecutive tokens that contain $\alpha$ bits of entropy, by our analysis of the distortion-freeness, each sampling of the next batch of tokens will have a uniformly random hash bit. Consequently, each sampling attempt will succeed in finding a consistent hash with probability $1/2$. After $\lambda$ attempts, our rejection sampling will find the next batch of tokens with probability $1 - 2^{-\lambda}$.

Additionally, if there exists a few $\ell$ consecutive tokens that does not contain enough entropy, our watermarking scheme can also handle these by error correction. Namely, our embedding algorithm will stop trying to embed the given bit at those locations and simply embed an arbitrary bit. As long as the number of such occurrences is fewer than $\gamma_{\max}$ the maximum number of errors that we can error-correct, the detection algorithm will still be able to recover the signature and output successful detection. $\square$

**Theorem 4.7** (Soundness). *$\mathcal{PDWS}$ is a $\ell$-sound publicly detectable watermarking scheme.*

*Proof.* The soundness of our watermarking scheme is based on the unforgeability of the signature scheme by a simple reduction.

If there exists a PPT adversary that can find a text labeled as watermarked, it must mean that this watermarked text has a valid message/signature pair embedded inside. Then, one may extract this pair, which constitutes a forgery attack against the underlying signature scheme. □

**Theorem 4.8** (Robustness). *$\mathcal{PDWS}$ is an $2(\ell + \ell \cdot \lambda_\sigma)$-robust publicly detectable watermarking scheme.*

*Proof.* The robustness of our scheme is rather easy to see. Let $\boldsymbol{t}$ be the output of the LLM. If the adversary's output $\mathcal{A}(\boldsymbol{t})$ contains $2\delta$ consecutive tokens from the original $\boldsymbol{t}$, it must mean that there is a $\delta$ consecutive tokens, which embeds a message/signature pair, is preserved in $\mathcal{A}(\boldsymbol{t})$. The detection algorithm will recover this consecutive sequence by an exhaustive search, resulting in a successful detection output. □

## 5 Empirical Evaluation

We implement both our publicly-detectable protocol and Christ et al. (2023)'s privately-detectable protocol—the only schemes with both cryptographic detectability and distortion-freeness at the time of writing. We focus our evaluation on assessing whether distortion-freeness is met in practice. In particular, we need to verify that our Assumption 3.1 on min-entropy is realistic. Note that our other formal properties, detectability and weak robustness, are immediate from our construction: detectability is inherited from the underlying signature scheme (BLS signaturesBoneh et al. (2001)), and weak robustness follows from the fact that the adversary does can only destroy all-but-one message-signature pairs, meaning that at least one message-signature pair is extractable at detection time. We additionally evaluate real-world performance under varying conditions. Concretely, we (a) present a range of generation examples for varying parameters in our protocol alongside examples from the other protocols, (b) quantify the distortion-freeness of the text completions using GPT-4 Turbo as a judge, (c) measure generation times against baseline (plain generation without any watermarking) and detection times for Christ et al. (2023) and our protocol, and (d) compare generation times for varying parameters in our protocol.

Hereafter, we will refer to the four generation algorithms using the following aliases:

1. **plain**. Standard text decoding.

2. **plain with bits**. Standard text decoding but with the arbitrary-to-binary vocabulary reduction of Christ et al. (2023) applied.

3. **symmetric**. The base (non-substring complete) version of the Christ et al. (2023) private key watermarking protocol.

4. **asymmetric**. Our public key watermarking protocol. We vary our protocol over three key parameters: signature segment length $\ell$, bit size $\beta$, and planted error limit $\gamma$.

Following prior watermarking evaluations, we use samples from the news-like subset of the C4 dateset (Raffel et al., 2020) as prompts. We implement our publicly detectable protocol and the base (non-substring-complete) version of the Christ et al. (2023) symmetric protocol. For `asymmetric`, cryptographic keys are sampled fresh for each parameter configuration, and we embed a single message-signature pair into the generated text. For `symmetric`, the key is fixed throughout. Our implementation is written in Python 3 with PyTorch (Paszke et al., 2019) and wraps around the Hugging Face `transformers` interface for transformer models (Face, 2023). We focus on the openly available Mistral 7B model (Jiang et al., 2023) for quality analysis. We additionally provide examples from the semi-open Llama 2 (Touvron et al., 2023) 70B and 13B models in Table 2 and Table 3, respectively. We use the 2.7B parameter OPT model (Zhang et al., 2022) for runtime analysis. Refer to Appendix C for extensive completion examples.

**Benchmarking procedure.** The benchmarking script selects a fixed number of prompts at random from the C4 dataset, skipping prompts that mention specific products. We run `asymmetric` first, then use the

number of tokens from the execution in the subsequent algorithms. This ensures that all algorithms produce the same number of tokens. We force generation length to be as long as needed to encode the signature, i.e., we explicitly block the model from outputting the stop token before the signature is embedded.

**Embedding in characters instead of tokens.** Note that throughout the paper we have discussed embedding the signature in *tokens* for simplicity and alignment with prior work. However, in our implementation, we plant the watermark directly on plain text rather than tokens to avoid inconsistencies in encoding then decoding (or vice versa) using any given tokenizer: tokenizers do not guarantee that encoding the same string twice will output the same tokens. Thus, the $\ell = 16$ and 32 in our subsequent discussion and figures will denote *characters*, not tokens. When $\ell = 32$ and $\beta = 2$, our gadget was embedded in $\approx 2,000$ tokens during the experiment. When $\ell = 16$ and $\beta = 1$ or 2, the gadget was embedded in $\approx 1,000$ tokens. For either case, $\lambda = 328$ or 360 bits of data were embedded depending on if $\gamma = 0$ or 2.

| # | Prompt | Plain (tokens) | Plain (bits) | Christ et al. | This work |
|---|--------|----------------|--------------|---------------|-----------|
| 1 | Windthorst pulled off a sweep of Collinsville Tuesday, while Archer City and Holliday were unable to advance.\nCHICO — With a chance to square off against the defending champs later this week, No. 7 Windthorst took care of business Tuesday night.\nThe Trojanettes | defeated Collinsville 25-16, 25-18, 23-25, 25-21 to secure an area title and punched their ticket to the Region I-2A quarterfinals with a trip to the controversial West Texas town of Iredell on the li | ,   seeded   second, swept   Collinsville   in three   games   to   set up   a   quarterfinal date   with   top-seeded Trent.   Windthorst defeated   Collinsville 25-16,   25-17   and 25-20.\nAssumption squeaked   past   Brock, 22-25, | swept Collinsville to advance to the area round of the postseason and face Olney.   They're hoping to turn the tables on the Ladycats for whom they lost in the second round a year ago.\n"They beat us la | traveled   to   No.   10 Collinsville and swept the Lady Collie Cardinals in game one, setting up a chance to take on Amarillo River Road later this week in a highly-anticipated Region I-2A area round rema |
| 2 | Eight Sussex skiers will take to the slopes to battle it out for the honour of being crowned National   Champion at the English Alpine Ski   Championships which start this weekend.\nBurgess   Hill sisters, 18-year-old Sam and 16-year-old Helen Todd-Saunders   and Crawley | /Fernhill   Heath   duo, Martin   Mellon   and Oliver James will don national kit to compete in Giant Slalom (GS) and Slalom (SL) events in Yongpyong,   South Korea.\nPerformances in 14 events will be ranked to | 's Amy Mertens, just 13, will go head-to-head with some of the country's best racers at the wide variety of disciplines on offer.\nWith almost 20 races in multiple disciplines it is the most diverse s | 's   Callum   Adams (18),   Ross   Guest (25)   and Max Green (17)   all   make   up   the squad for the championships.\nWorthing's Danny Williams (17), Syd Wilson (16) and East Grinstead's Naomi Wilkinson (21) also re | 's   Amy Crocket   will all   compete   in   the Senior National Championships, racing at the Trois Vallees ski resort in   the   French   Alps, along   with   Booker's Mollie   Darling   and Dylan   Jetsun;   Rye's William Phil |

Table 1: Example completions from Mistral 7B (Jiang et al., 2023). For the Christ et al. (2023) scheme, we set security parameter $\lambda = 16$. For our scheme, we set signature segment length $\ell = 16$, bit size $b = 2$, and maximum number of planted errors $\gamma = 2$. Completions are truncated to the first 200 characters. See Table 5 for more completions under the same conditions.

## 5.1 Text Completion Examples

We show how text completions vary over six benchmarking runs with different generation parameters. We primarily use the Mistral 7B model due to its high quality output for its size class. We display a couple text completions for each algorithm in Table 1. See Table 4 through Table 9 in Appendix C for the full collection of text completions—each table shows one text completion per generation algorithm for 5 distinct prompts. We additionally include a few completion examples from larger models (Llama 2 70B and 13B) in Table 2 and Table 3. In the next section, we discuss the quality of these examples.

## 5.2 Zero-Shot Quality Measurements with GPT-4 Turbo

Following many works in the NLP literature (e.g., (Chen et al., 2023; Piet et al., 2023), we automatically assign a quality score to each text completion using an established LLM. We do not use model perplexity as it is known to assign arbitrary scores in some cases—for example, it can favor repetitive text (Holtzman et al., 2019; Welleck et al., 2019; Piet et al., 2023). In particular, we use zero-shot prompting of GPT-4

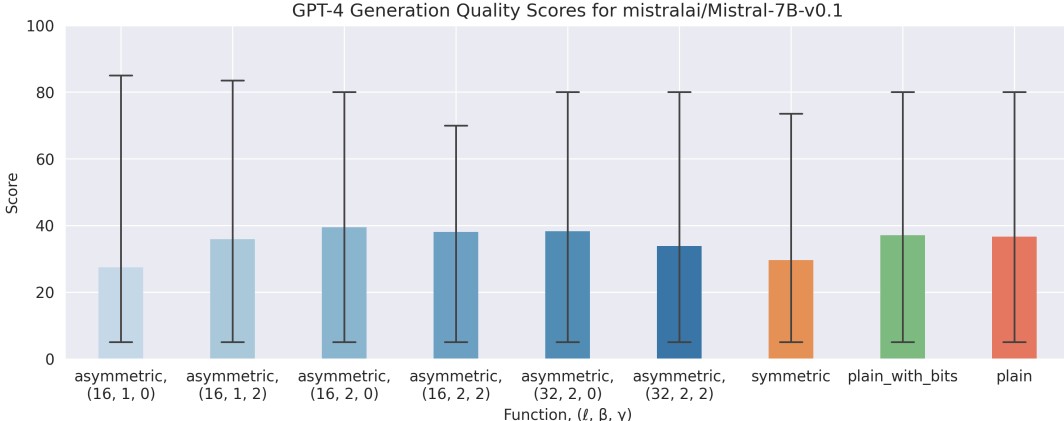

Figure 2: Aggregated text quality score assignments from GPT-4 Turbo for each generation algorithm configuration over the Mistral 7B model (Jiang et al., 2023). For asymmetric, the configurations from left to right represent the most compact (lowest quality) to least compact (highest quality) parameters. Each bar is the aggregation of GPT-4 Turbo-assigned quality scores for 250 distinct prompt completions. The error bars show the 95% interval data spread. Observe that no protocol clearly outperforms the others: the mean score falls between 27 and 40 for all protocols, and each one exhibits large quality spreads. Note that even the baseline decoder, plain, follows this pattern. This suggests the watermarking protocols are indeed distortion-free.

Turbo (OpenAI, 2023). For each batch of four generations (one from each algorithm), our prompt template asks the model to: (a) rate the text completion by giving it a score from 0 (worst) to 100 (best), and (b) give reasoning for the assigned score in list form.

In theory, all the algorithms should be computationally distortion-free if their underlying assumptions are satisfied. Recall distorion-free means no PPT algorithm can distinguish between watermarked and non-watermarked text. We see in Figure 2 that GPT-4 Turbo-assigned scores have similar means and high variance—there is no statistically-significant signal that any particular generation algorithm outperforms the others. This provides evidence toward real-world distortion-freeness.

**On embedding compactness.** One of the main limitations of our protocol is that it takes a relatively large number of characters (tokens) to embed the message-signature pair (greater than 1,000 tokens for our most compact parameter configuration where $\ell = 16$, $\beta = 2$). Our GPT-4 Turbo quality scores are comparable to the plain baseline even for the most compact parameters, suggesting that we can encode more information in even less tokens at the cost of increased runtime. That is, we can decrease $\ell$ (holding $\beta$ constant) or increase $\beta$ (holding $\ell$ constant).

## 5.3 Generation and Detection Runtimes

In this section, we discuss the generation and detection runtimes shown in Figure 3.

**Text generation.** plain generation without any watermarking or bit reduction is, as expected, the fastest—we use this setting as our control against which we compare the performance of the watermarking schemes. plain with bits and symmetric are closely correlated, implying that the dominating cost of Christ et al. (2023)'s watermarking scheme is the arbitrary-to-binary vocabulary reduction. The asymmetric scheme ran approximately twice as fast on each prompt for the parameters we used in this experiment.

**Watermark detection.** We see asymmetric detection runs significantly faster than symmetric detection. asymmetric consistently runs near 0.01s and symmetric varies between 10 and 10,000s for Mistral 7B. This aligns with performance expectations. Detecting an asymmetric watermark takes constant time in our implementation because we know the starting index of the signature. Note that in extensions of this implementation where the location of the signature in the text is unknown, detecting an asymmetric watermark

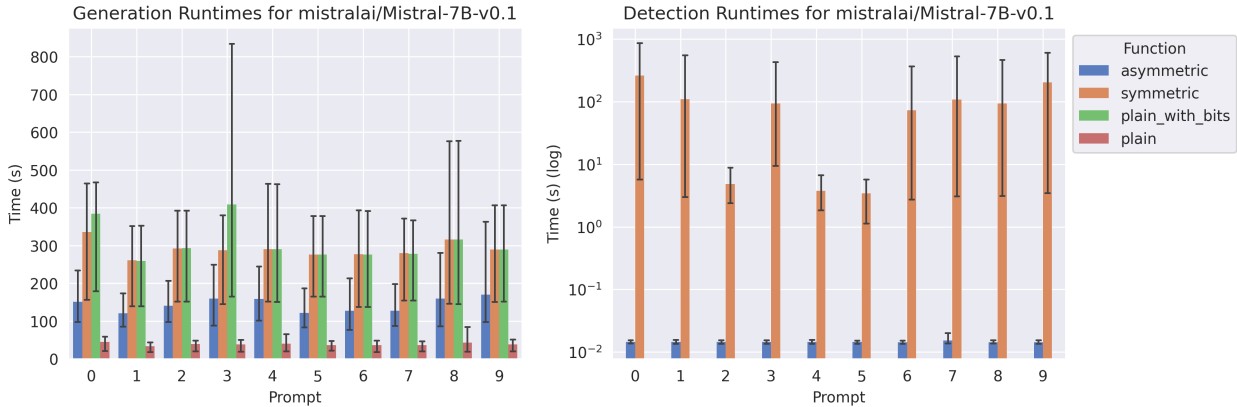

Figure 3: Generation and detection runtimes for each generation algorithm over the Mistral 7B model (Jiang et al., 2023). Five distinct generations were aggregated for each of the 10 random prompts from the news-like portion of the C4 dataset (Raffel et al., 2020). The error bars show the 95% interval data spread. On average, the fastest to slowest generation runtimes were for: plain (expected as this is the baseline), asymmetric, then symmetric and plain with bits (the latter two are about equal with the dominant cost being the reduction to a binary vocabulary). For detection, asymmetric runs much faster than symmetric which is expected given they run in linear vs. quadratic time, respectively, in the number of tokens $n$.

would take linear time in the generated text length $n$ using a sliding window (see Appendix B.1). On the other hand, detecting a symmetric watermark runs in quadratic time based on the generated text length $n$ (Christ et al., 2023).

## 5.4 Asymmetric Parameters Optimized for Runtime

Expected generation time is proportional to the average number of characters needed to encode the watermark: $E_\lambda(\ell, \beta) = 2^\beta \cdot \frac{\lambda}{\beta} \cdot \ell$, where $2^\beta$ is the expected number of attempts to pass the rejection sampling, $\frac{\lambda}{\beta}$ the number of signature segments required, and $\ell$ is the number of characters per signature segment. Holding all else constant, observe that (a) higher $\ell$ means more entropic flexibility but increases runtime by a linear factor, and (b) higher $\beta$ also means more entropic flexibility but increases runtime by a factor of $\frac{2^\beta}{\beta}$.

We see in Figure 4 that the empirical results line up with the expectations above when comparing the runtimes across differing parameters. Runtimes for $\ell = 16, \beta = 1$ and $\ell = 16, \beta = 2$ are close to each other and approximately double the runtimes for $\ell = 32, \beta = 2$.

## 5.5 The Effect of Error-Correction

Figure 4 presents the generation time performance of our publicly-detectable protocol for six different parameter settings. We see that across the board, allowing the algorithm to plant up to $\gamma = 2$ errors resolves high runtime spread. Generations that took a relatively long time got stuck in the rejection sampling loop trying to find a hash collision. The version of our protocol that plants errors was designed to allow the algorithm to break out of this loop by settling for the closest hash (as measured by Hamming distance to the target bit sequence). In particular, we see significant reductions in runtime spread for prompts 4, 5, 6, 8, and 9.

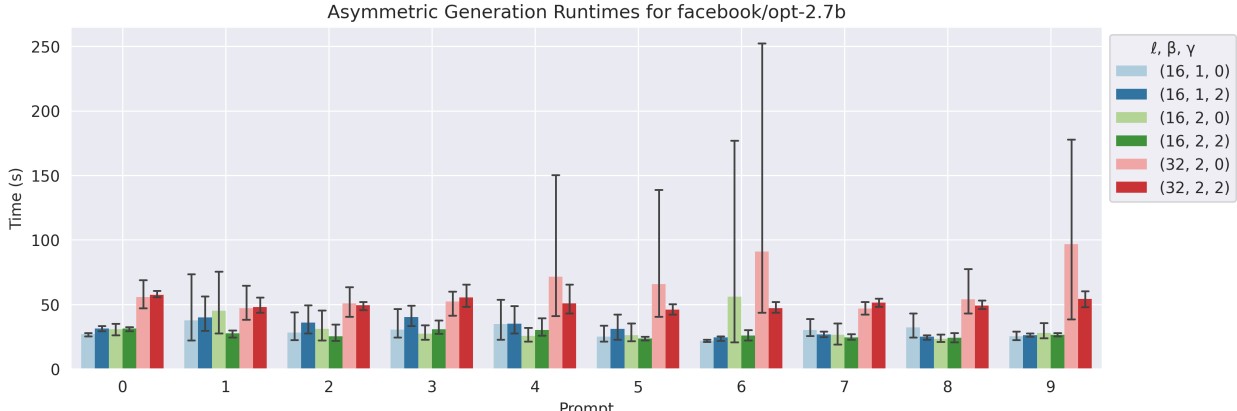

Figure 4: Generation runtimes for each variant of our protocol over the OPT-2.7B model (Zhang et al., 2022). Generation runtimes for different parameter instantiations of our protocol. 5 completions were generated for each of the 10 random prompts from the news-like portion of the C4 dataset (Raffel et al., 2020). $\ell$ denotes the signature segment length, $\beta$ denotes the bit size, and $\gamma$ denotes the maximum number of planted errors. The error bars show the 95% interval data spread. Comparing non-error-corrected ($\gamma = 0$) vs. error-corrected ($\gamma = 2$) runtimes for prompts with high variance (prompts 4, 5, 6, 8, and 9 for parameters $\ell = 32$, $\beta = 2$ and $\ell = 16$, $\beta = 2$), we can see a clear reduction in the variance and mean runtime when error correction is applied to overcome low entropy periods. Note that we expect to sample $E_\lambda(\ell, \beta) = 2^\beta \cdot \frac{\lambda}{\beta} \cdot \ell$ characters to embed the signature codeword. Thus, in expectation, $E_\lambda(16, 1) = E_\lambda(16, 2) < E_\lambda(32, 2)$ where $\lambda = 328$ or 360 depending on if $\gamma = 0$ or 2. Our empirical runtimes align with this.

# 6 Related Work

## 6.1 Text Distinguishers

We discuss key approaches for detecting AI-generated text without introducing any changes to text generation. See (Jawahar et al., 2020) for a comprehensive survey.

Early approaches to detecting AI-generated text revolve around looking for features of AI-generated text that are not present in human-generated text—if you can or cannot identify such features, you can conclude the text was or was not AI-generated. Examples of features include relative entropy scoring (Lavergne et al., 2008), perplexity (Beresneva, 2016), and other statistical signals (Gehrmann et al., 2019). We refer the reader to (Beresneva, 2016) for a survey.

Another common method is to train another model to automatically identify distinguishing features. Research of this nature (Zellers et al., 2019; Mitchell et al., 2023; GPTZero, 2023; Hendrik Kirchner et al., 2023) uses deep learning as a binary classifier.

The problem with this idea is that it relies on AI-generated text being fundamentally different from human-generated text. This reliance is at odds with the core goal of LMs: to produce human-like text. As models get better, statistical features of AI-generated text will decay. In particular, GPT-4 (OpenAI, 2023) and other cutting edge models are quickly closing this gap. Chakraborty et al. (2023) formally show that as AI-generated text approaches human quality, text distinguishers demand longer text samples.

Beyond relying on an diminishing assumption, text distinguishers lack formal guarantees—the detector's correctness is only empirically validated, so any validation performed is only relevant to the exact model, its configuration, and prompting context during experimentation.

Other work has shown that it is possible to train models to transform text such that it fools text distinguishers (Krishna et al., 2023; Sadasivan et al., 2023).

## 6.2 Watermarking Schemes

There is a recent line of work using ML to perform watermarking (Abdelnabi & Fritz, 2021; Qiang et al., 2023; Yoo et al., 2023; Munyer & Zhong, 2023; Liu et al., 2023). Notably, Liu et al. (2023) address the same problem as this paper: their approach is to train two models—one for embedding a signal and one for detecting it. This is analogous to using asymmetric keys. Crucially, all schemes in this category are entirely empirical and have no formal guarantees such as correctness, soundness, or distortion-freeness.

Recently, Kirchenbauer et al. (2023) gave the first watermarking scheme with formal guarantees. They showed that when model entropy is high, a watermark can be planted by hashing previous tokens to embed a watermark signal in the next token. Crucially, hashing tokens to zero or one effectively assigns a binary label to potential next tokens. By ensuring that only tokens with label "zero" appear in generated text, the watermark can be detected after text generation by recomputing the hash. Kirchenbauer et al. (2023) bound the distortions introduced by the watermark by measuring perplexity: the difference between the distribution produced by the plain model and the one produced by the model with watermarking.

The Gumbel softmax scheme of Aaronson (2023) is another approach to LLM watermarking. The scheme uses exponential minimum sampling to sample from the model using randomness based on previous tokens (via hashing). This scheme is distortion-free so long as no two output texts that share a common substring are public (Christ et al., 2023). This is unlikely for a widely-used LLM.

Kuditipudi et al. (2023) design a family of watermarking schemes that aim to maximize robustness. The main idea of their scheme is to use a key that is as large as the generated text output—this permits statistical distortion-freeness as opposed to the cryptographic distortion-freeness of this paper and Christ et al. (2023) at the cost of computation that scales with the generated text output. The long key is then "aligned" with the generated text by computing an alignment cost—this alignment cost can be (re)computed at detection time with the detection key and the generated text. Text that has been watermarked will be statistically likely to have low alignment cost at detection time. Their scheme has the desirable property that the alignment cost is a measure of edit distance and thus a watermark may persist even if text is inserted or deleted from the original watermarked text. Their scheme is not provably complete or sound.

We remark that prior watermarking schemes differ in trade-offs when token length increases (or decreases). Schemes in the private key setting ((Kirchenbauer et al., 2023; Christ et al., 2023; Kuditipudi et al., 2023)) gain stronger soundness as token length increases and vice versa. In this work, strong soundness is achieved so long as there are sufficiently many tokens to embed our message-signature gadget, i.e., there is a sharp threshold. This is because soundness in our scheme stems from unforgeability of the signature, rather than strength of an embedded statistical signal.

Piet et al. (2023) systematically analyze watermarking schemes in the secret key setting. Their study focuses on assessing generation quality and robustness to minor edits for practical protocol parameters. They state that the Kirchenbauer et al. (2023) scheme produces the best watermark even though the protocol is distortion inducing. Furthermore, they conclude that distortion-freeness is too strong a property for practice. This conclusion was drawn from quality assessment performed by the chat version of Llama 2 7B (Touvron et al., 2023). We remark that Llama 2 7B's quality assessment likely does not generalize—higher fidelity models may reveal weaknesses in distortion-inducing watermarking schemes. In contrast, no (probabilistic, polynomial time) algorithm can distinguish between non-watermarked text and distortion-free watermarked text so long as protocol assumptions hold.

Zhang et al. (2023) formally proved that "strong" robustness is impossible in watermarking schemes. The further demonstrated that their attack works in practice against a range of secret-key watermarking schemes (including the Kuditipudi et al. (2023) scheme). That is, it is possible to remove watermarks with low computational effort whilst preserving text quality. Our scheme comes under their "weak watermarking scheme" definition and thus their impossibility result does not apply.

Qu et al. (2024) developed a watermarking scheme for LLMs that also makes use of ECC. They use ECC to gain robustness: this is distinct from this work, which uses ECC to overcome low entropy periods when generating text.

### 6.3 Linguistic Steganography

Linguistic steganography generalizes LLM watermarking. The main goal is to embed a hidden message in natural language text. A steganographic protocol provides formal security if an adversary cannot determine whether a message is from the original distribution or the distorted distribution that embeds a hidden message (Hopper et al., 2002). The key difference in this setting compared to watermarking is that distortions to the distribution are permitted so long as some notion of semantic similarity is preserved. Furthermore, there are critical differences in the problem model between linguistic steganography and LLM watermarking. In LLM watermarking, prompts are adversarially chosen and the watermarking protocol should be agnostic to the plain text distribution. The focus of linguistic steganography is to achieve undetectability. In LLM watermarking, undetectability is not important—what is important is that the text is of similar (ideally, the same) quality as unwatermarked text, i.e., it should be distortion-free. That is, watermarked text should still be usable for the same downstream tasks for which unwatermarked text is useful.

### Broader Impact Statement

Detecting AI-generated content is of paramount importance. For large language models, watermarking has emerged as a promising approach. While there are prior solutions for watermarking language models, our paper resolves a fundamental problem with prior watermarking schemes—public verifiability. Namely, our protocol uses different keys for watermarking and detection, meaning that anyone can detect the presence of a watermark without surrendering the watermark key (which can be nefariously used to planting watermarks in arbitrary text). Our scheme has direct impact and real world applicability for cases where watermarked text does not need to be strongly robust against edits. One such example is if the output of a RAG model is only provided to customers that are incentivized to keep the text intact.

On the technical side, our work is a significant departure from all existing private-verifiable watermarking schemes. In particular, our work employs rejection sampling to embed a publicly verifiable signatures to the output of the language model. This serves as a first step in this direction for the general AI-generated content detection problem. We believe that the rejection sampling approach is amenable to robustness improvements through novel use of error-correction schemes that are resilient to insertions and deletions.

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

## A  Additional Preliminary

Our proof relies on the following standard concentration bound, known as Hoeffding inequality.

**Theorem A.1** ((Hoeffding, 1994))**.** *Let $X_1, \ldots, X_n$ be independent random variables such that $a_i \leq X_i \leq b_i$ for all $i$. Let $S_n = \sum_{i=1}^{n} X_i$. For any $t > 0$, it holds that*

$$\Pr\left[|S_n - \mathrm{E}\left[S_n\right]| \geq t \cdot \mathrm{E}\left[S_n\right]\right] \leq 2 \cdot e^{-\frac{2t^2}{\sum_{i=1}^{n}(b_i - a_i)^2}}.$$

## B Extra Discussion

### B.1 Minor Protocol Extensions

There are a number of extensions one can add to the protocol to make well-defined improvements with no significant cost.

**Arbitrary length output**  In the technical overview, the detector crucially relies on knowing the precise position in the text where the message and signature is embedded. We can relax this by searching over all windows of length $\ell$ for the message in linear time—it must be that the corresponding signature will be embedded in the next $\ell \cdot \lambda_\sigma$ tokens after the $\ell$ message tokens. Furthermore, in order to generate $n$ tokens, we tile multiple message-signature pairs until enough tokens are generated. If $n$ is not divisible by $\ell + \ell \cdot \lambda_\sigma$, we generate the remaining tokens using the native model sampler.

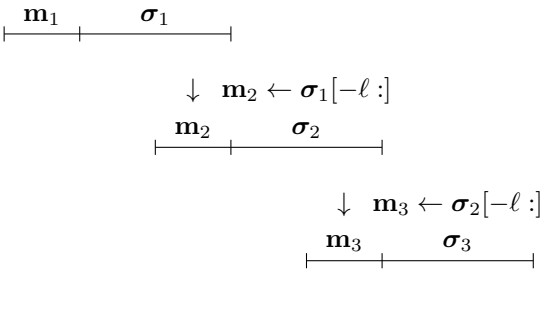

Figure 5:  Tiling structure to compress multiple message-signature pairs. This is possible because the signature itself is pseudorandom.

**Embedding multiple watermarked blocks**  The scheme described above embeds one signature in a fixed number of output bits. To extend this scheme to support arbitrarily large output, we can tile the block structure defined above sequentially until the desired length is reached.

When $n$ is large enough to permit multiple message-signature pairs, we can leverage the pseudorandomness of the signature to use significantly fewer tokens. Specifically, to embed $k$ message-signature segments, we only need $k \cdot (\ell + \ell \cdot \lambda_\sigma) - (k-1) \cdot \ell$ tokens given $\lambda_\sigma \geq \ell$. Note that in practice, $\lambda_\sigma = 328$ bits (inherited from BLS signatures (Boneh et al., 2001)) and $\ell \simeq 16$ characters. The idea is to use the last $\ell$ bits of the signature from the previous message-signature pair as the message for the next segment. This preserves full distortion-freeness due to the pseudorandomness of the signature scheme. See Figure 5 for a visual description of this process.

**Tuning the robustness vs. distortion-freeness trade-off**  We assume that each block of $\ell$ tokens has sufficient entropy (as defined by the parameter $\alpha$) to ensure our formal notion of distortion-freeness is met. However, in practice, one can set $\ell$ to a concrete value (e.g., determined empirically for a specific model and hyperparameters) to tweak robustness—$\ell$ should be set as low as possible such that the text quality remains similar[3] to non-watermarked text in order to get more robustness. When $\ell$ is low, each message-signature segment requires fewer total tokens, meaning more segments can be embedded in $n$ tokens. So long as one message-signature pair remains after text edits, the watermark is detectable.

**Chaining embedded bits**  The plain gadget embeds one bit of the signature into a block of $\ell$ tokens. Observe that the scheme is susceptible to the following attack. If an adversary knows that a specific portion of text corresponds to a message-signature pair, she can construct a different message that still verifies. By definition of the random oracle, any freshly sampled token has $1/2$ probability of hashing to 0 or 1. The

---

[3]Our Theorem 4.5 proves that this similarity is closely related to how much entropy each $\ell$ tokens carry.

adversary replaces a signature $\boldsymbol{\sigma} \in \mathcal{T}^{\ell \cdot \lambda_\sigma}$ with a new message $\boldsymbol{\sigma}' \in \mathcal{T}^{\ell \cdot \lambda_\sigma}$ by changing $\ell$ tokens at a time and checking that the new block still hashes to the same value as the old block. If the hash is inconsistent, sample a new block of $\ell$ tokens and try again. This process can be repeated for all $\lambda_\sigma$ bits of the signature. In addition, since the blocks are independent, the adversary can replace each block with adversarial text in a modular fashion.

We make this attack more difficult by introducing dependencies across adjacent blocks. For each $i \in \lambda_\sigma$ success condition for rejection hashing changes from

$$H\left(s_{i,1}^{\sigma_{i,1}} \cdots s_{i,\ell}^{\sigma_{i,\ell}}\right) = \sigma_i$$

to

$$H\left(\bigoplus_{j=1}^{i} s_{j,1}^{\sigma_{j,1}} \cdots s_{j,\ell}^{\sigma_{j,\ell}}\right) = \sigma_i$$

where $\bigoplus_{j=1}^{i} v_j$ stands for concatenation as $\bigoplus_{j=1}^{i} v_j := v_1 \parallel v_2 \parallel \ldots \parallel v_i$. Now, for the adversary to perform the same signature replacement attack, she can no longer change each $\ell$-length token block independently.

**Dynamic entropy** Through this paper, we assume that there is sufficient entropy in every $\ell$ tokens sampled from the LLM in order to simplify presentation and analysis. We can relax this assumption in the real world—rather than setting a global length $\ell$ that is expected to be of sufficient entropy, we can empirically measure how much entropy is available from the LLM at generation time. This idea originates from Christ et al. (2023) where they use it to transform their private-key scheme into a substring-complete version, i.e., a version that embeds independent detectable segments to gain robustness. We can employ the core idea in the asymmetric setting as follows:

Given a distribution $p \leftarrow \mathsf{Model}(\cdot, \cdot)$, one can measure the entropy of sampling token $t$ from $p$ as $-\log(p(t))$. Thus, each time a new token is sampled, the generation algorithm can keep track of how much collective entropy has been seen up to that point, i.e., cumulative entropy can be measured as $\sum_{i=1}^{j} -\log(p_i(t_i))$ for a contiguous block of $j$ tokens—this sum can be updated incrementally as more tokens are sampled. To use this optimization, the generation algorithm simply waits for sufficiently many token samples such that the cumulative entropy sum is large enough. Once the threshold is reached, all sampled tokens are taken as the message, and the protocol proceeds as before[4]. At detection time, the message is no longer of fixed size, so it no longer suffices to iterate all windows of fixed length in the text. Thus, the detector will incur quadratic performance costs by searching over all possible message strings from the input text.

## B.2 Outsourcing Watermarking Itself

One of the main benefits of a publicly detectable watermarking scheme is that the watermark detector is outsourceable—the entity providing a "detection service" is different from the one providing the model. Besides outsourcing detection, we remark that our protocol also naturally supports outsourcing the watermarking process itself, i.e., an *external entity* can embed a publicly detectable watermark in text generated by a *private model*. In contrast with all known prior watermarking schemes, our protocol does not necessarily need to know the distribution from which to sample each token—it suffices to obtain a list of "next best tokens" given by the prompt and prior generated tokens. This means that our watermarking scheme can operate over API access to private LMs.

Assume that an LLM provider supports an API that returns the top $\ell$ next tokens for a given prompt. The watermarking process itself can be outsourced by replacing direct model calls with API calls. This has the downside of requiring linearly many requests in the output length—we acknowledge that this is likely a preventative expense for many use cases.[5]

---

[4]Note that this optimization can only apply to the message portion of the embedding—applying it to the signature component would result in an exponential blowup of the detector's runtime.

[5]At the time of writing, OpenAI charges US\$0.03 per 1000 input tokens and US\$0.06 per 1000 output tokens for GPT-4 API access with 8k token context.

### B.3 Performance Optimizations

The main computational bottleneck in our scheme is rejection sampling. There are two straightforward optimizations to this process that would greatly improve concrete performance. First, rejection sampling naturally lends itself to parallelism. Instead of sequentially searching for a signature collision (on the CPU), this process can be performed in parallel on the GPU to take advantage of (a) faster inference and (b) faster hashing. Second, consider the case where $\ell = 1$ for simplicity. Whenever a token is rejected (i.e., the token hash did not match the signature hash), this information can be used to modify model logits to prevent sampling the same token repeatedly. This has a large impact on the expected running time of rejection sampling since if the token was sampled, it is likely to be sampled again. Preventing this situation drastically improves the time needed to find a matching token. This idea can be generalized to $\ell > 1$, however, care would be necessary to ensure that only the specific token sequence becomes improbable after each rejection.

### B.4 Embedding Codewords for Robustness

Unlike prior watermarking schemes (Kirchenbauer et al., 2023; Christ et al., 2023; Kuditipudi et al., 2023), our framework of *embedding extractable bits into the generated text* readily allows for further improvement in robustness.

Instead of embedding a message-signature pair (which contains no redundancy) into the generated text, one may embed an "error-corrected encoding" of the message and signature into the generated text. Given an input text, the detection algorithm will first extract the potentially-erroneous codeword from the text and then apply error correction to recover the original message-signature pair. Apparently, the robustness of this new watermarking scheme will inherit the robustness of the error-correction scheme. For instance, if the error-correcting algorithm allows for 10% of errors, our watermarking scheme will be resilient to 10% of word replacements.

We emphasize that the robustness guarantee provided in this scheme can be *formally proven*, unlike the robustness claims in prior works, which are based on experimental results and heuristics.

One potential barrier to this proposed scheme is efficiency. The efficiency of the scheme depends on the efficiency of the error-correcting encoding. Normally, in the context of text editing, one aims for resilience against insertions and deletions. However, existing state-of-the-art error-correcting codes against insertions and deletions (Guruswami & Wang, 2017) have worse efficiency compared to their counterparts for Hamming error correction. However, we emphasize that our framework is modular. Any improvement in the construction of error-correcting codes will directly give improvement to the efficiency of this proposed scheme.

## C Completion Examples

Our evaluation primarily involved the Mistral 7B (Jiang et al., 2023) completions discussed above, we ran the following benchmarks on the larger Llama 2 13B and 70B models (Touvron et al., 2023) on a smaller scale. All model inference was performed with full precision—no quantization was employed. Evaluation was computed using NVIDIA A100 GPUs with 40GB VRAM (from 1 to 8 in parallel depending on the model size) for all benchmarks except those involving OPT 2.7B, which ran on a single NVIDIA RTX 3090 Ti GPU.

| # | Prompt | Plain (tokens) | Plain (bits) | Christ et al. | This work |
|---|--------|----------------|--------------|---------------|-----------|
| 1 | Windthorst pulled off a sweep of Collinsville Tuesday, while Archer City and Holliday were unable to advance.\nCHICO — With a chance to square off against the defending champs later this week, No. 7 Windthorst took care of business Tuesday night.\nThe Trojanettes | held off a pesky Chico squad en route to a 64-48 victory in the Region II-2A girls basketball semifinals.\nNow they'll get a rematch with No. 1 Cooper in the region title game.\nWindthorst is medal-bo | swept No. 21 Collinsville 25-19, 25-19, 25-19 to win their Class 2A Region II quarterfinal at Chico High School.\nWindthorst (26-9) advances to region semifinals that will be held in Dangerfield Frida | won behind 19 kills from Audrey Lopez to take Game 1, 25-14, 25-17, 25-21, over Collinsville in the Region II-2A quarterfinal match, setting up a semifinal tilt against Crawford for Friday afternoon i | unleashed some offense and fended off a Bearcats' comeback attempt to finish off a 67-58 win and sweep of Collinsville in the Class 2A girls area playoffs.\n"I'd like to see us play with that type of |
| 2 | Eight Sussex skiers will take to the slopes to battle it out for the honour of being crowned National Champion at the English Alpine Ski Championships which start this weekend.\nBurgess Hill sisters, 18-year-old Sam and 16-year-old Helen Todd-Saunders and Crawley | 's Olivia Gillespie, 17, will race in the girls' events at at Whakapapa Ski Area in New Zealand.\nThe boys competing for the honour of becoming National Champions are 19-year-old Ivan Ukri, of Coldean | 's Nicholas Moynihan, 21, are part of the 11-strong Team Evil who head to Bormio, Italy, tomorrow (Tuesday) for a week's training ahead of the championships. The trio will be joined by three other Loo | -based Sol Buchler (17) are travelling out to Montalbert, France for the Championships which run from March 21st to March 24th due to lack of snow conditions in the UK.\nSam Todd-Saunders. Pic by Acti | 's Trevor McColgan have all previously won national titles and will be looking to continue their winning ways.\nHowever, they will be up against a strong field of competitors, before one girl and one |

Table 2: Example completions from Llama 2 70B (Touvron et al., 2023). For the Christ et al. (2023) scheme, we set security parameter $\lambda = 16$. For our scheme, we set signature segment length $\ell = 16$, bit size $b = 1$, and maximum number of planted errors $\gamma = 2$. Completions are truncated to the first 200 characters.

| # | Prompt | Plain (tokens) | Plain (bits) | Christ et al. | This work |
|---|--------|----------------|--------------|---------------|-----------|
| 1 | Windthorst pulled off a sweep of Collinsville Tuesday, while Archer City and Holliday were unable to advance.\nCHICO — With a chance to square off against the defending champs later this week, No. 7 Windthorst took care of business Tuesday night.\nThe Trojanettes | downed No. 10 Collinsville 75-27 in a Class (1A) bi-district basketball playoff game Tuesday as part of a quadruple-combo event at Chico High School.\nWindthorst (22-8) jumped out to an 11-4 lead afte | (19-7-2) defeated Collinsville, 6-3, in the Region I-1A semifinals at Lion Field. They will face Honey Grove, 27-4, in the regional final at Noon Friday in Stephenville.\nNo. 5 Honey Grove defeated Va | beat the Class 2A Region I power Collinsville 4-2 Tuesday at the Big Country Soccer Complex.\nIt was one of four games in the opening round of the Region I-2A tournament, with just three wins enough t | stayed on course for another appearance in Friday's Class 1A state championship, pulling off a 3-1 sweep of Collinsville to advance to the state tournament semifinals.\n"I think the main thing that h |
| 2 | Eight Sussex skiers will take to the slopes to battle it out for the honour of being crowned National Champion at the English Alpine Ski Championships which start this weekend.\nBurgess Hill sisters, 18-year-old Sam and 16-year-old Helen Todd-Saunders and Crawley | siblings, 18-year-old Ollie O'Sullivan and 14-year-old Jack, are among those earmarked to shine in the event based at Essex Snowsports.\nTheir handler, Alton snow sports coach, Liz Baird said: "It's a | 's Amy Conroy (pictured), 18, will represent Sussex against some of the fastest alpine racing female representatives in the country.\nJohn Gardner, father of 19-year-old Eliza and Tom, both pictured, | 's 14-year-old Helen Cullen all enjoy supporting each other at competitions and training.\nBoth Cullen and Todd-Saunders looked on in admiration as Sam won the Salomon super-g last season snatching vi | brothers Harry Roylance, 16, and Charlie Roylance, 13 of Oxted's College Sevenoaks have all been selected to head off to the slopes of Austria for the competition.\nThe event includes a full programm |

Table 3: Example completions from Llama 2 13B (Touvron et al., 2023). For the Christ et al. (2023) scheme, we set security parameter $\lambda = 16$. For our scheme, we set signature segment length $\ell = 16$, bit size $b = 1$, and maximum number of planted errors $\gamma = 2$. Completions are truncated to the first 200 characters.

| # | Prompt | Plain (tokens) | Plain (bits) | Christ et al. | This work |
|---|--------|----------------|--------------|---------------|-----------|
| 1 | Windthorst pulled off a sweep of Collinsville Tuesday, while Archer City and Holliday were unable to advance.\nCHICO — With a chance to square off against the defending champs later this week, No. 7 Windthorst took care of business Tuesday night.\nThe Trojanettes | beat Collinsville, 6-1, 7-0, to secure a season sweep of the former Division I champs and advance to the Region II-1A quarterfinals later this week.\n"I thought, going into this tournament, one of the | knocked off Collinsville in two five-set matches to win a Class 1A bi-district volleyball playoff series.\nWindthorst is scheduled to visit fourth-ranked Mark, a fellow District 8-2A squad, Friday at | opened play Tuesday in the Class A Region I 1A regional quarterfinals with a tough 25-20, 25-20, 25-22 sweep of Collinsville.\n Unfortunately for the Trojans and No. 13 Archer City, Westbrook beat the | pulled off a sweep at Lamar Nice Field as they handled No. 17 Collinsville in the Area Championship, winning 10-0 in five innings and 5-1 in a very tightly contested second game.\nWindthorst will fac |
| 2 | Eight Sussex skiers will take to the slopes to battle it out for the honour of being crowned National Champion at the English Alpine Ski Championships which start this weekend.\nBurgess Hill sisters, 18-year-old Sam and 16-year-old Helen Todd-Saunders and Crawley | gymnast Paige McKenzie will represent England at girls combined category whilst Crawley siblings Philippa and James Horton will compete in the boys hopefuls event.\nSteyning pair Phoebe Pratten and La | students, 20-year-old Alice Reidy, 18-year-old Felix Rogers, 16-year-old Tom Eagleson and 15-year-old Patrick Lane, all racers for the Brighton Snozone Academy, will all be looking to grab the title. | skier, 14-year-old Joe Roberts will be competing in the event in Meribel, France, from 5th March.\nThe rest of the team is made up of Alexandra Leff (Worthing), Jack Hatton, Phoebe Smith (Both 18, Has | Girls' School's 17-year-old Lucy Finlayson, who is also the under-21 British Slalom Champion, will go head to head with some of the best skiers in the country in a series of races taking place on Ste |
| 3 | It's 14 degrees and snowing. But at one of Moscow's new cooperative clothing markets, business is booming. Muffled against the cold, vendors shout promotions for their paltry offerings while others mingle more discreetly with the crowd, hawking French perfume | , jewelry, and other just-arrived wares, all at a price.\n"Get a look at these earrings!" shouts one such trader, whipping in a customer with his pendulum-like swings of his color-plastic earrings.\ | and Italian earrings.\n\nIt's not all as innocent as it sounds. All of the clothing are stolen from retailers and vendors have to be careful because many of them are not only police but private detect | and Jean-Paul Gaultier sweaters under the table.\n\nThe business may be illicit, but the scene is all too familiar in Eastern Europe's newest attempt at capitalism: the Communist free market.\n\n"I m | , Chinese sneakers and Swedish sweaters. With cash prizes up for grabs for the best deals, this is Russian capitalism at its most jaded. "We only go to places like that now," a university vice-chancel |
| 4 | Now Finally Taking Shape: A World With Ever Less Decently-Paid Work for Most. But Must This Be a Problem?\nFor centuries, experts/futurists have predicted that machines would someday make workers obsolete. And now it's finally happening, sporadically, here and | there. But is this anything to worry about? Is this a problem? The root of these hard-fought disagreements about it is the unspoken assumption that (at least) the working class is supposed to comprise | there,\nBut let's be honest here – pre-retirees who know there's little hope that they'll be able to fill large gaps in the decades remaining before & after retirement – they don't know what to do f | there, little by little. And while it isn't much of a shock to most of us to hear that sophisticated algorithms and machines are better than us at many types of fairly routine work (or that sexy/futur | there–the interesting structural aspect is that the service sector (low-tech), not the high-tech sector is leading the charge.\nAs I outlined in the first installment, the jobless-growth paradox is |
| 5 | In our "always on" culture, the office mantra is: work late, stay connected. The problem is that working harder and longer doesn't necessarily make you more productive. Research shows that getting away—especially on a journey that engages your mind and body, | not just your eyes—is essential to achieve powerful rebooting and offer deep drives toward your own life goals.\n\nYou will need a "model of journeying," wholly embracing the unfolding holiday experie | slows you down, and helps you reset—revitalizes and energizes.\n\n>Get out of your rut – find the ultimate Wilderness Lodge Villa Package, and escape to paradise.\n\nWalt Disney World Resort vacation | maybe even encourages you to step outside your comfort zone—can spur productivity, creative thinking, and innovation.\nBy Leo Babauta of Zen Habits\n\n"Be here now." This isn't a sentence that needs | and forces you to unplug—can encourage your productivity when you get back.\nIn fact, nothing nurtures the drive to excel more than seeing your passion played out in a different context—say, by exp |

Table 4: Example completions from Mistral 7B (Jiang et al., 2023). For the Christ et al. (2023) scheme, we set security parameter $\lambda = 16$. For our scheme, we set signature segment length $\ell = 16$, bit size $b = 2$, and maximum number of planted errors $\gamma = 0$. Completions are truncated to the first 200 characters.

| # | Prompt | Plain (tokens) | Plain (bits) | Christ et al. | This work |
|---|--------|----------------|--------------|---------------|-----------|
| 1 | Windthorst pulled off a sweep of Collinsville Tuesday, while Archer City and Holliday were unable to advance.\nCHICO — With a chance to square off against the defending champs later this week, No. 7 Windthorst took care of business Tuesday night.\nThe Trojanettes | defeated Collinsville 25-16, 25-18, 23-25, 25-21 to secure an area title and punched their ticket to the Region I-2A quarterfinals with a trip to the controversial West Texas town of Iredell on the li | , seeded second, swept Collinsville in three games to set up a quarterfinal date with top-seeded Trent. Windthorst defeated Collinsville 25-16, 25-17 and 25-20.\nAssumption squeaked past Brock, 22-25, | swept Collinsville to advance to the area round of the postseason and face Olney. They're hoping to turn the tables on the Ladycats for whom they lost in the second round a year ago.\n"They beat us la | traveled to No. 10 Collinsville and swept the Lady Collie Cardinals in game one, setting up a chance to take on Amarillo River Road later this week in a highly-anticipated Region I-2A area round rema |
| 2 | Eight Sussex skiers will take to the slopes to battle it out for the honour of being crowned National Champion at the English Alpine Ski Championships which start this weekend.\nBurgess Hill sisters, 18-year-old Sam and 16-year-old Helen Todd-Saunders and Crawley | /Fernhill Heath duo, Martin Mellon and Oliver James will don national kit to compete in Giant Slalom (GS) and Slalom (SL) events in Yongpyong, South Korea.\nPerformances in 14 events will be ranked to | 's Amy Mertens, just 13, will go head-to-head with some of the country's best racers at the wide variety of disciplines on offer.\nWith almost 20 races in multiple disciplines it is the most diverse s | 's Callum Adams (18), Ross Guest (25) and Max Green (17) all make up the squad for the championships.\nWorthing's Danny Williams (17), Syd Wilson (16) and East Grinstead's Naomi Wilkinson (21) also re | 's Amy Crocket will all compete in the Senior National Championships, racing at the Trois Vallees ski resort in the French Alps, along with Booker's Mollie Darling and Dylan Jetsun; Rye's William Phil |
| 3 | It's 14 degrees and snowing. But at one of Moscow's new cooperative clothing markets, business is booming. Muffled against the cold, vendors shout promotions for their paltry offerings while others mingle more discreetly with the crowd, hawking French perfume | and warm Palestinian wraps. – Peggy Randall, Moscow resident MILAN - The past 20 years of Italian history have come violently to life as I began reading Turino: la storia nel presente ("Turin: Histor | .\n\nSome sellers have finally located a rare commodity: bras for small women, as well as perfume. (One fragrance is dubbed "Happiness.")\n\nThe market in Beschastovo, northeast Moscow, is the latest | , Greek olive oil, Italian cognac.\n\nThis hush-hush fare all has one thing in common: It is on sale outside the official government quota for foreign-currency imports. It is on the black market.\n\nH | and Japanese ski suits - both sold in rubles at local markets only a few years ago.\n\nClothing experienced a boom in summer 1990, when new supplies arrived in retail markets priced in hard currency |
| 4 | Now Finally Taking Shape: A World With Ever Less Decently-Paid Work for Most. But Must This Be a Problem?\nFor centuries, experts/futurists have predicted that machines would someday make workers obsolete. And now it's finally happening, sporadically, here and | there–perhaps five percent of workers at any given time–and it's contributing to the unemployment and social breakdown that's getting increasing media attention. But do we have a reason to think thi | there. But many people don't want to admit that it is happening. First Software, Next Robots.\n\nBut the soon-to-be-likely reality is that many jobs will be automated and many people will be happier a | there. Is this a disaster to be avoided–or welcomed?\n\n#367 #use-Canada As Canada uncovers 88 secret RCAF bases that housed 15,973 MIAs. Fact— all of them captives of the Secret Occupation Governm | there in this major trend: The Destruction of Work/Jobs. Though machines might take jobs/work away, what if these inventions are not "the end", but the hinge, the pivot of a new era? A period of less |
| 5 | In our "always on" culture, the office mantra is: work late, stay connected. The problem is that working harder and longer doesn't necessarily make you more productive. Research shows that getting away—especially on a journey that engages your mind and body, | experiences new people and cultures, and captures inspiring sights, sounds and smells, rejuvenates you, making you more productive and creative when you return. Around the world, travel planners and h | perhaps in creating new curiosities–can be the key to productivity. One of the ways to do that is to go immersed into a New Place.\n\nWith offices in Hong Kong, New York and San Francisco, Signature J | and takes you far away from your daily schedule—achieves a greater reset than going on vacation.\n\nGlobe-trotting can positively impact creativity as well. Divergent thinking is an essential part of | and promotes creative connections—delivers what McKinsey Co. CEO Doonan Stewart calls "the invaluable renewal to our psyche and brains."\n\n### Section 5: Customer Research\n\nBeen with my current cu |

Table 5: Example completions from Mistral 7B (Jiang et al., 2023). For the Christ et al. (2023) scheme, we set security parameter $\lambda = 16$. For our scheme, we set signature segment length $\ell = 16$, bit size $b = 2$, and maximum number of planted errors $\gamma = 2$. Completions are truncated to the first 200 characters.

| # | Prompt | Plain (tokens) | Plain (bits) | Christ et al. | This work |
|---|--------|----------------|--------------|---------------|-----------|
| 1 | Windthorst pulled off a sweep of Collinsville Tuesday, while Archer City and Holliday were unable to advance.\nCHICO — With a chance to square off against the defending champs later this week, No. 7 Windthorst took care of business Tuesday night.\nThe Trojanettes | advanced to the Region I-2A final by sweeping Collinsville, 25-12, 25-11, 25-13, at Chico High School. The win sets up a four-match all-Vernon District series against No. 2 Brock that will decide who | rolled to a 35-11 victory over Collinsville in the area round of the play-offs, setting up a date with No. 3 Parsons in the regional quarter-finals at 7 p.m. Friday in Archer City.\nThat is when postsea | pulled off a sweep of Collinsville, 25-18, 25-13, 25-22 to move into Thursday's Region I-2A final against Post (24-15), the defending Region I-2A champion.\nThe winner faces the Region II-2A winner on | took a large halftime lead and were never really threatened in a 58-46 victory at Collinsville in the title game at Freer's Hall.\nWindthorst will travel to Bellmead this weekend to take on District |
| 2 | Eight Sussex skiers will take to the slopes to battle it out for the honour of being crowned National Champion at the English Alpine Ski Championships which start this weekend.\nBurgess Hill sisters, 18-year-old Sam and 16-year-old Helen Todd-Saunders and Crawley | dynamo, 17-year-old Hamish Lovegrove will compete against 214 other skiers in Busillats to be crowned National Champion.\nBritish development squad member, Isaac Brown and 15-year-old Amber Pyrah, fro | teenager Bradley Leech have made it through to finals in Bansko, Bulgaria.\nThe trio, who represent Mid Sussex Ski Racing Club, are among 119 university students and amateurs taking part in events tha | 's 18-year-old Henry Rees, will compete in the giant slalom on the Da Jaunne slope at the Chaudanne in Les Arcs, whilst 14-year-old Sol Steed (Chichester) will compete in Downhill 1 and Yusuf Sardar S | team-mates, 16-year-old Jimmy Simpkin and 17-year-old Maria Brooksbank lead the way in the teenage categories, but they face competition from many of the country's best racers over the four day event |
| 3 | It's 14 degrees and snowing. But at one of Moscow's new cooperative clothing markets, business is booming. Muffled against the cold, vendors shout promotions for their paltry offerings while others mingle more discreetly with the crowd, hawking French perfume | and Italian shoes they bought in Paris. Each takes the risk of negotiation as the market's secondary vendors only accept rubles for merchandise that is against the law to sell. Little returns to these | and women's lingerie.\n\nOn a market wall, multi-colored textured bags emblazoned with the trademark of a young designer line the length of a refrigerator. A vendor tells me that a woman sold 30 of th | and knockoff designer clothes. They have been lured here by better money.\n\nThursday should be a holiday for the people of Caucasus, Georgia's breakaway republic of Abkhazia, Abkhazia has a war-damag | and Japanese crystal.\n\nRussian President Boris N. Yeltsin's effort this year to reshuffle the Kremlin and the government and to draft the nation's new national budget have led to a blistering shake |
| 4 | Now Finally Taking Shape: A World With Ever Less Decently-Paid Work for Most. But Must This Be a Problem?\nFor centuries, experts/futurists have predicted that machines would someday make workers obsolete. And now it's finally happening, sporadically, here and | there–and in splurgy 1st-world nations like Canada, it's getting harder and harder to get a job and if you're lucky enough to get one, the pay is lousy.\nRobots Start Stealing Jobs From Immigrants | there. Politicians and the media repeatedly show hosts and studio personnel working on futuristic-looking self-driving cars that need no human driver, and other vehicles taking us short distances in d | there. So while some still cry 'it can't happen' or 'but it will create so much wealth for everyone'/'shouldn't we try to extend the benefits of computers/i-tech to as many people world-wide as possibl | there. But is this really a problem?\n\nEconomics/class/globalization:\nThe Role of the US Border Patrol in the Racial Inequality Plaguing the San Diego Region. Black and Latino areas have significan |
| 5 | In our "always on" culture, the office mantra is: work late, stay connected. The problem is that working harder and longer doesn't necessarily make you more productive. Research shows that getting away—especially on a journey that engages your mind and body, | with long-term health benefits and a chance to nurture your creativity—can boost productivity and enhance your career. Nearly three out of four (74%) of workers believe spending time on vacation insid | and has a clear achievement in sight—helps us attain productivity, greater mental clarity, and a renewed sense of self.\n\nMore than a vacation, expedition experiences provide a way for your employees | improves your memory, boosts creativity, and fosters innovation. Whether traveling is for business or leisure, there are ways to incorporate fitness and mindfulness into your time away.\n\n# Why Fitne | like with a ski trip—can help us better understand who we are and where we want to go next.\n\nHere at the Aspen Journalism Podcast, we can't emphasize the necessity of unplugging enough. Whether you |

Table 6: Example completions from Mistral 7B (Jiang et al., 2023). For the Christ et al. (2023) scheme, we set security parameter $\lambda = 16$. For our scheme, we set signature segment length $\ell = 16$, bit size $b = 1$, and maximum number of planted errors $\gamma = 0$. Completions are truncated to the first 200 characters.

| # | Prompt | Plain (tokens) | Plain (bits) | Christ et al. | This work |
|---|--------|----------------|--------------|---------------|-----------|
| 1 | Windthorst pulled off a sweep of Collinsville Tuesday, while Archer City and Holliday were unable to advance.\nCHICO — With a chance to square off against the defending champs later this week, No. 7 Windthorst took care of business Tuesday night.\nThe Trojanettes | swept past Collinsville 25-20, 25-17, 25-20 in the Region I-2A second-round playoff match at Chico High School. Windthorst (32-16) earned the 25-point victory with two buzzer-beating kills that fell s | swept Collinsville in the Region II-A quarterfinals, beating the Lady Lions, 25-15, 25-21, 25-23. After two straight must-win, Region II-B semifinal matches, Archer City and Holliday were both ousted | defeated Collinsville 58-57 in Game 1 of the Class 2A Regional I final doubleheader at Chico on Tuesday and swept the pivotal series with a 56-45 win in Game 2.\nIt was a hilariously close finish to t | swept Collinsville 25-13 and 25-15 on Tuesday, advancing to a rematch with No. 2 Divine Child Academy.\n\nThe two teams met in group play last week at the Windthorst tournament, with DCA winning both |
| 2 | Eight Sussex skiers will take to the slopes to battle it out for the honour of being crowned National Champion at the English Alpine Ski Championships which start this weekend.\nBurgess Hill sisters, 18-year-old Sam and 16-year-old Helen Todd-Saunders and Crawley | 's 17-year-old Freddie Nairn head to Alta Badia, Italy along with former Welsh international Mackenzie Hughes, 22, London Welsh Hilary Grant, 14, London skiing superstar Grace Coombs, aged 22, and Max | 's Anugreen Sefi are all set to compete as part of a 100 strong English team which will be aiming for glory at Pralognan-la-Vanoise.\nSnow-lovers started their journey to the French resort of the way | 's Sophie Smith are all in the GS Ladies' Under 19s while Guin Bacon, from Eastbourne, races in the GS Women's Under 21s.\nThe final English ladies and men's squads for the upcoming FIS World Junior C | 's girls' team will spearhead the Brighton & Hove City Ski Team (BHCST) hopes, with the cross country team also likely to figure prominently.\nSam, a former pupil at Burgess Hill School for Girls who |
| 3 | It's 14 degrees and snowing. But at one of Moscow's new cooperative clothing markets, business is booming. Muffled against the cold, vendors shout promotions for their paltry offerings while others mingle more discreetly with the crowd, hawking French perfume | , unregistered guns of various awkward calibers or pills which might be anything from cranberry antioxidants to Viagra.\n\nAfter the Soviet Union collapsed in 1991, the Soviet economy slipped into a d | and luxury watches at discount prices.\n\nWelcome to the well-heeled world of "gray" market, which both provides much needed relief for Russia's overtaxed consumer sector and spotlights major ineffici | and knockoff handbags to the agog shoppers. Official statistics show that half of all Russians now live chronically on the brink of poverty– with an income well below the minimum clothing needs for th | and German refrigerators.\n\nDespite an all-out advertising storm, which covered apartment walls with huge lecture posters proclaiming productivity and dignity through hard work, numerous Soviet trad |
| 4 | Now Finally Taking Shape: A World With Ever Less Decently-Paid Work for Most. But Must This Be a Problem?\nFor centuries, experts/futurists have predicted that machines would someday make workers obsolete. And now it's finally happening, sporadically, here and | there, largely because of the Internet. And here at Atrios, a group of somewhat leftish Americans have welcomed this development. But why?\n\nFor a long time these pundits have worried that, either in | there, as it always will increasingly - except, unfortunately, the experts/futurists didn't think to anticipate that machines would also be the solution to their forecasted problem.\n\nUnfortunately, | there, and many observers fear that the economic future is bleak. We are already seeing this with automation (i.e., robots) replacing workers in retail jobs, warehousing jobs, manufacturing jobs in pl | there, and that means more job losses/underpay/zero-hour-or-part-time work for most. It doesn't exactly feel like a triumph. So at least some experts worry that white collar knowledge jobs are also c |
| 5 | In our "always on" culture, the office mantra is: work late, stay connected. The problem is that working harder and longer doesn't necessarily make you more productive. Research shows that getting away—especially on a journey that engages your mind and body, | and encourages a more expansive thinking—is enormously invigorating. These are the kinds of journeys JTB Special Interest Groups offer.\n\n### We Have Fun:\nWe offer numerous ways to connect with fe | not just your brain—increases innovation and creativity, which leads to better performance when workers return, according to studies cited by the World Travel and Tourism Council.\nEscapes that comb | and provides ample time to think—fuels your brain to solve complicated problems. So, while productivity may not look productive, your brain will be working overtime during your downtime.\n\nFind a Gea | such as a bicycle tour—can improve your performance when you return.\n\nCycling opens up your imagination by focusing your attention on one of the most fundamental human activities—moving. With no ga |

Table 7: Example completions from Mistral 7B (Jiang et al., 2023). For the Christ et al. (2023) scheme, we set security parameter $\lambda = 16$. For our scheme, we set signature segment length $\ell = 16$, bit size $b = 1$, and maximum number of planted errors $\gamma = 2$. Completions are truncated to the first 200 characters.

| # | Prompt | Plain (tokens) | Plain (bits) | Christ et al. | This work |
|---|--------|----------------|--------------|---------------|-----------|
| 1 | Windthorst pulled off a sweep of Collinsville Tuesday, while Archer City and Holliday were unable to advance.\nCHICO — With a chance to square off against the defending champs later this week, No. 7 Windthorst took care of business Tuesday night.\nThe Trojanettes | crushed No. 27 Collinsville in a sweep at Mineral Wells High School, 25-10, 25-18, 25-22. However, controversy preceded the final scores.\nAfter the first game, the Collinsville coach took umbrage wit | beat up on Angelina County's Collinsville, 25-10, 25-15, at Pike Provident Bank Gym as the only Nolan County School District No. 12 team still alive in the Region I-2A tourney.\nMPVP MARLI HOUSER spen | swept Holliday in two games, winning by the scores of 25-9, 25-12 to move on to Thursday's regional semifinal game in China Spring against the winner of Brookshire Royal and Waxahachie Life Christian. | swept the Collinsville Lady Lions 25-15, 25-16, 25-16 in the Region II-2A playoffs to advance to the area round. Windthorst was the only team in District 9-2A to beat defending state champion Friona |
| 2 | Eight Sussex skiers will take to the slopes to battle it out for the honour of being crowned National Champion at the English Alpine Ski Championships which start this weekend.\nBurgess Hill sisters, 18-year-old Sam and 16-year-old Helen Todd-Saunders and Crawley | '-based Louise D'Arcy, will be among the youngest in a field comprised of some of the best junior slalom skiers from across the UK.\n\nThe Championships take place over four days at the Les 2 Alpes ski | 's Karl Holland (17) will compete over a six day period in Tignes, France, braving the 'weapons of mass destruction' that France has become famous for.\nLaura Rees (20 – Poole), Alabaster Jones (17 – | sisters Ella and Imogen Kowal are joined by 15 year-olds Sophie Gwarunski from Burgess Hill and Chichester's Lizzie Kuzmenko and for-times National Champion Rebecca Burns, who is 17 and from Lewes.\nF | 's Nancy Webb, 13, Rudi Hudman, 15, Jack Auger, 16 and Ben Harsant, 20 and 19-year-old Harrogate skier Lucy Try, will compete in England's largest ski race of the season which starts at the Alpe d'Hue |
| 3 | It's 14 degrees and snowing. But at one of Moscow's new cooperative clothing markets, business is booming. Muffled against the cold, vendors shout promotions for their paltry offerings while others mingle more discreetly with the crowd, hawking French perfume | and expensive cosmetics.\n\nThey defy the law. underneath the table trade is illegal, but the bearded young man swiftly managing the exotic goods, Darwin, has nothing to fear. He's a policeman.\nDar | and fake Rolex watches rather than the cheap fleeces and caps that dominate the market.\n\nOn a clear day, when the snow lies slushy on the sidewalk, patrolling cops evict the successful and respected | packaged in Russian boxes.\n\n## Survivors Of Morocco Gas Plane Crash Feted In Paris\nDecember 03, 2012\nhttp://www.huffingtonpost.com/2012/11/30/morocco-gas-plane-crash_n_2212279.html\nParis, Fra | , Italian pens and English socks. On the fringe of this human maze three young women show off a collection of sherry-brown cotton skirts, dotted with small medallions. Ostentatiously, they all have "b |
| 4 | Now Finally Taking Shape: A World With Ever Less Decently-Paid Work for Most. But Must This Be a Problem?\nFor centuries, experts/futurists have predicted that machines would someday make workers obsolete. And now it's finally happening, sporadically, here and | there, in small bites, and unpredictably. But it's truly hard to even imagine what this will mean for our future, barring unexpected large-scale changes in attitudes about the nature of work and in ou | there, in certain fields. As noted elsewhere many times over the years here, we're seeing this mostly already in high-margin/sexy fields first (as in finance, movie industry, design) and continuing to | there, as an increasing percentage of workers are losing their jobs to machines, and to corporate greed.\n\nIs this really a problem? TaskRabbit CEO says not, that there's no class war, and he's hopin | there, but still mostly in traditional manufacturing. ('You Build It, We Just Take the Money': Afraid of your resumÁ©? Apply at McDonald's!: Is work becoming a dying institution? Who actually works t |
| 5 | In our "always on" culture, the office mantra is: work late, stay connected. The problem is that working harder and longer doesn't necessarily make you more productive. Research shows that getting away—especially on a journey that engages your mind and body, | such as a Disney vacation— can actually relieve the symptoms of stress.\n\nWhat better place to reconnect with your "inner you" than the peace, pampering and productivity of a Walt Disney World® vacat | one that unplugs you from your daily routine—can bring real benefits that last long after crusty sandals have been discarded.\n\n"Travel is about having a different experience," says Dr. John Pencavel | challenges you to push beyond your comfort zone—can help you recover from burnout and make you a more effective, productive worker. Work breaks are increasingly popular, giving people an opportunity t | and provides a change of scenery—can leave you more energized and better able to focus when you return.\n\n### Passages Anamcara Mini Retreat\n\nThese five-hour luxurious retreats offer an exquisite |

Table 8: Example completions from Mistral 7B (Jiang et al., 2023). For the Christ et al. (2023) scheme, we set security parameter $\lambda = 16$. For our scheme, we set signature segment length $\ell = 32$, bit size $b = 2$, and maximum number of planted errors $\gamma = 0$. Completions are truncated to the first 200 characters.

| # | Prompt | Plain (tokens) | Plain (bits) | Christ et al. | This work |
|---|--------|----------------|--------------|---------------|-----------|
| 1 | Windthorst pulled off a sweep of Collinsville Tuesday, while Archer City and Holliday were unable to advance.\nCHICO — With a chance to square off against the defending champs later this week, No. 7 Windthorst took care of business Tuesday night.\nThe Trojanettes | clipped the No. 25 Lady Panthers of Collinsville in three sets, but in quite a display. The Girls finished the night 25-17, 25-11 and 25-8.\nWas the display impressive? Well, let's just say it was the | made quick work of Chico, sweeping the Wildcats easily in the opening round of the Chico district tournament.\nIn the Class A championship today at Chico High School, Windy will face Petrolia, who ups | swept their Class 1A Region II Area rival Collinsville, 25-16, 25-11, 25-19 to advance to the regional quarterfinals and pull a step closer to a rematch with defending state champion and area foe Chil | , victorious over Collinsville 3-0, will play No. 2 MLK in the Region I-3A volleyball semifinals at 12:30 p.m. Friday in Austin.\nThe Lady Saints were upended 3-1 by Wellington in Tuesday's remaining |
| 2 | Eight Sussex skiers will take to the slopes to battle it out for the honour of being crowned National Champion at the English Alpine Ski Championships which start this weekend.\nBurgess Hill sisters, 18-year-old Sam and 16-year-old Helen Todd-Saunders and Crawley | 's 20-year-old rower turned skier Cameron Bourke, are all determined to do the club proud.\nThe Nationals are a sell-out event attracting athletes from all over the country and sees around 400 competi | twins, 18-year-old sisters, Ellie and Rachael Bell are all racing at the English Alpine Ski Championships over the next two weeks hoping to take home two crystal ski trophies to represent Southern and | 's Jake Moffat (17) have all been selected to compete in the opening races to decide the top five challenged for 2014.\nThe squads selected to join the three young slalom and giant slalom skiers are: | downhill skier Francesca Poulton will compete with the national team in the Under 21 categories.\nNigel Thompson, coach of Birchwood Ski Club in Brighton which develops skiers, said: "We are really p |
| 3 | It's 14 degrees and snowing. But at one of Moscow's new cooperative clothing markets, business is booming. Muffled against the cold, vendors shout promotions for their paltry offerings while others mingle more discreetly with the crowd, hawking French perfume | (1,500 rubles a bottle), stolen army uniforms (50 rubles) and Chinese-made T-shirts topped with knock-off designer logos (4 rubles each).\n\n"It's good business," says Elena Gumerova, who runs the sta | or fake designer jeans. In a tent filled with suitcases piled with handbags, watches and bottles of perfume, Russian, Chinese and East European women buy.\n\nMany of the items are fake. Even this most | , Japanese designer suits and the finest Hungarian chocolates. (This is typically a big week for Moscow retailers, as Russian women usually wait until the December 7th Day of Reconciliation and the Ne | and jeans. "GreatBart knows his stuff–the thunder roll will be the guitar hook. And the chorus turns it into one great one-word question. - RoatMarked I'll. But the cold snap is threatening to pierc |
| 4 | Now Finally Taking Shape: A World With Ever Less Decently-Paid Work for Most. But Must This Be a Problem?\nFor centuries, experts/futurists have predicted that machines would someday make workers obsolete. And now it's finally happening, sporadically, here and | there, and particularly in developed nations. Yesterday Switzerland became the first employer of an army of robots (geolocated in the nation's food courts and brick-and-mortar retail outlets), the fir | there, and the general public is not yet at all ready to cope with it. Humans are being pushed out of ever more work. Africans too have already found their minds and fists no match for automated produ | there in earnest, and more often in job casualization and gigification.\nHere in North America and western Europe we're seeing tens of millions of jobs just vanish or be photographed or programmed or | there. But what does this say about progress– and the ability of most to keep from falling permanently into the ranks of the unemployed and marginally-employed?\nSky-High Marginal Taxes Are Pushed by |
| 5 | In our "always on" culture, the office mantra is: work late, stay connected. The problem is that working harder and longer doesn't necessarily make you more productive. Research shows that getting away—especially on a journey that engages your mind and body, | helps to sharpen focus, relax and reset. Outside the hotel lobby and chain restaurants, the waterfront and hiking trails of this historic city invite connection and encourage a fresh perspective. Here | enhances your creativity, and revitalizes your energy—puts you on the path toward sustained success.\n\nVirgin's Branson Group is offering leadership development trips to exotic locations that break o | whether it's a retreat in the desert or a weekend on the water —creates a personal and professional recharge.\n\nHow? You pause and get perspective on where you're at, where you're going, and what mat | like a climbing trip—can spur your creativity, stoke your innovation, and make you feel productive and fulfilled. By taking the trip, you recharge emotionally, physically, and even cognitively. Yes, |

Table 9: Example completions from Mistral 7B (Jiang et al., 2023). For the Christ et al. (2023) scheme, we set security parameter $\lambda = 16$. For our scheme, we set signature segment length $\ell = 32$, bit size $b = 2$, and maximum number of planted errors $\gamma = 2$. Completions are truncated to the first 200 characters.

