# OpenReview forum: "Publicly-Detectable Watermarking for Language Models"
_TMLR — Rejected by TMLR_

### Review · Reviewer_F5qw · 2024-06-17

**Summary Of Contributions:**

The submission proposes an asymmetric watermarking scheme for LLM that is
- distortion-free (high quality of generated text),
- complete (watermarked detected if text long enough),
- sound (an attacker cannot watermark without the secret key).

**Audience:**

No

**Broader Impact Concerns:**

NtD

**Claims And Evidence:**

No

**Requested Changes:**

# Ambiguity
There is clearly an ambiguity about the functionality of the watermark scheme.
The cited works deal with robust watermarking for the detection of AI-generated text.
This work is about fragile watermarking for text integrity.

If the authors still pretend to propose a robust watermarking scheme, then they must compare runtimes, text length, and robustness to Aaronson, Kudipiti, etc . Their scheme will be miserable, yet the only one being sound and complete, but so unrealistic.

The only way to go is to take the other direction: this is fragile watermarking offering a different functionality than these other schemes.
This is somewhat already stated ("*Our scheme has direct impact ... for cases where watermarked text does not need to be strongly robust*", "*an entity may want to use the watermark as a signature for their content*") but very hidden in the text. I request the term **fragile** or **integrity verification** to appear in the title and abstract and that the introduction make it clear that this paper is about robust watermarking.

# Counterfeit attack
- The counterfeit attack is real. It should be discussed in the main body of the paper. The threat model of Def. 3.4 excludes this attack but it is so unrealistic.

# Experimental benchmark
- the examples of completions are misleading since they are truncated.
- the poor quality of the text in Fig. 2 deserves more precautions. The score for the "plain" LLM (without watermark) is measured on truncated texts? texts as long as the watermarked texts? or plain text without any constraint (ie. much shorter)?

# Minor comments
- Introducing  3 hash functions may help understanding (instead of re-using H_1).
- I have the impression that the text describes a very simplified version of the algorithm that doesn't match the algorithms. On the other hand, I have the impression that Appendix B.1 proposes extensions already described in the algorithms (but not the main body of text).
- Are you sure Line 7, Algo 4 matches with Line 7, Algo. 5? Why including m and c_prev in H_1 (Line 7, Algo. 4)?
- Is H_2 really needed? Are the bits of the ECC codewords really biased?

**Strengths And Weaknesses:**

# Weaknesses

The main weakness is that this paper was submitted to a previous conference, where it received high-quality reviews. I have difficulty finding any substantial differences between the two versions. This shows a lack of respect for the work of the previous reviewers. Like their watermark scheme (Algo. 4), there is a rejection sampling: keep submitting until acceptance.

To summarize the previous reviews:
- Lack of "watermarkability": the generated text must be very long (~ thousand tokens) to convey the watermark signal. Indeed, the watermark embedding forces the LLM to produce long text. Moreover, the runtime is huge: several tens of seconds for one text generation (see Fig. 4).
- Absolutely no robustness (not a single character modification)... or, more precisely, a non-realistic assumption: if the attacker doesn't change ~2,000 consecutive tokens, then the scheme is robust.
One counterfeit attack was disclosed. This pulls down Th. 4.7, whose proof falls short: the scheme is as sound as the digital signature ... except that it relies on other bricks (LLM) not as sound. Appendix B. 1 seems to propose a patch, but I believe the attack is still pending. Just resample the text sequentially.
- No citation to similar ideas in the watermarking literature known as fragile or semi-fragile watermarking.

I would personally add:
- This paper should be submitted to a cryptography journal. There is nothing related to machine learning. It uses an LLM as a cryptographic primitive described in Def. 3.5. As a machine learner, I have difficulty reading this cryptographic style, for instance, the equation of Def. 3.6. This is out of the scope of TMLR.
- I see the importance of distortion-freeness. I don't see the point of undetectability. In the listed applications, anyone knows the output text is watermarked. You are confusing with LLM steganography. Moreover "*without the secret watermarking key, no PPT machine can distinguish plain LLM output from watermarked LLM output*". The public key detector is a PPT machine.  Another one: output text strangely too long -> watermarked.

---

> ### Author Response · Authors · 2024-08-01
> **On suitability, model weakness, and robustness**
>
> **Why is our paper interesting for ML venues?** There is no doubt that the question of watermarking language models is a major question for machine learning and has drawn much attention recently. In this realm, we are the first that study it in the public detection setting and are the current state-of-the-art. Achieving public detection is a highly sought-after property for watermarking schemes and we see our paper as a first step toward this important research direction. Although our paper is cryptography-heavy, this does not make it less interesting for the ML audience. In fact, many papers accepted at ML conferences/journals are very cryptography-heavy. For example, the Christ, Gunn and Zamir scheme [2], another cryptographic watermarking scheme to appear at COLT 2024, is a pure crypto theory paper. Furthermore, watermarking is not the only machine learning-related problem that cryptographic techniques can be applied to. The FairProof paper [3] was accepted at ICML 2024 and received the best paper award at the Privacy-ILR workshop. These are just a few recent examples and our paper is written in the same cryptographic style as these examples.
>
> **On the "main weakness."** We incorporated all constructive feedback. Along with this repeat review, the paper received high-quality reviews which were addressed in the submitted version. The first review, which gave our paper a rating of 7: accept, asked us to clarify the extent to which real-world attacks are covered by our proofs. We assured the reviewer that our evaluation sufficiently covered all claimed properties, which the reviewer thanked us for. The second high-quality review described two weaknesses of our model, claiming that the model was too weak for real-world use (these concerns were reproduced above). In rebuttal, we argued that our scheme is parameterizable and the proposed attacks weaken as the watermarked text length increases. Long-form watermarked text is becoming more relevant as more models support large token output, so we believe that this is within reason. We updated the paper directly in response to this on page 5 in the “On the unforgeability of our scheme” paragraph. Moreover, the only other existing work in the public-detection setting is susceptible to similar attacks [5]. We do not agree with the claim that we ignored the reviewer’s constructive feedback.
>
> **Lack of robustness.** As referenced in the paper, no watermarking scheme can achieve robustness in the strongest sense. Generic attacks were formalized by Zhang et al. [1], and even before then, robustness was already questionable due to simple attacks like roundtrip translation, interleaving, etc. Therefore, in this work, we aim for a weak form of robustness where the watermark is preserved as long as a sequence of consecutive tokens of a certain length is copied. This weaker form of robustness is still highly desirable for several reasons. Firstly, unlike the empirical (strong) robustness claimed by many prior works, which are eventually compromised, as shown by Zhang et al. [1], our approach provides provable robustness. Additionally, it comes with other provable guarantees, such as distortion-freeness and unforgeability, which earlier works do not achieve. Secondly, this limitation in robustness is not unique to our scheme. All existing watermarking schemes that offer provable guarantees can only achieve this level of (provable) robustness. Improving robustness is indeed an exciting direction for future research efforts. In fact, a recent paper by Golowich and Moitra [4] studies this exact problem. Finally, this level of robustness is still meaningful in many practical scenarios. Consider a student using ChatGPT to write an essay. While a student could potentially avoid plagiarism detection by rephrasing every other sentence, such a level of modification requires substantial effort and alters the text enough that arguably it no longer constitutes straightforward plagiarism. This demonstrates that even our weaker robustness can effectively deter and identify significant copying of content.

---

> > ### Author Response · Authors · 2024-08-01
> > **On distortion-freeness/undetectability and fragile/semi-fragile watermarking**
> >
> > **Distortion-freeness vs. Undetectability.** First, in our paper, we only require our scheme to be distortion-free and do not require it to be undetectable. Second, a certain level of undetectability (in the following sense) is implied by achieving distortion-freeness. Namely, if one generically transforms a (unwatermarked) LM into a watermarked one in a distortion-free manner, it has to be the case that one cannot detect if a given generated text is coming from the unwatermarked model or watermarked one. In other words, any way of detecting if a text is watermarked or not does break distortion-freeness.
> >
> > **On citing semi-fragile/fragile watermarking prior work.** We follow the framework from Zhang et al. [1]. They introduce two classes: strong robustness (which they show does not exist) and weak robustness (which we use in our paper). In the line of work on watermarking for applications such as copyright (i.e., watermarking before the emergence of LLMs), robust, semi-fragile, and fragile refer to the following: a robust watermark preserves the watermark under a range of transformations, but it cannot pinpoint if an image with a detectable watermark is the original image or not. Semi-fragile and fragile watermarking are intended to protect the integrity (and authenticity) of the image at the cost of robustness to varying degrees. I.e., robustness is weakened in order to allow for some ability to pinpoint which parts of the image have changed. Our protocol does not do this: its function is to provide very strong publicly-verifiable authenticity (i.e., false positives will not occur). Of course, we too trade off robustness for these guarantees. Thus, we use the reasonable terminology of Zhang et al. [1]. We are happy to add citations for papers on semi-fragile/fragile watermarking as it is indeed related work.

---

> > > ### Comment · Reviewer_F5qw · 2024-09-20
> > >
> > > The authors and I disagree... once again. I acknowledge that the technicity of the paper is very good. Yet, it is a pity that the authors continue presenting their work as a robust watermarking technique whereas it is not at all robust. The proposed technique can not be used for generated text detection since modifying a single token deludes the watermark detection. Moreover, the technique applies only to very long generations.
> > >
> > > The proposed scheme is indeed a fragile watermark that protects the integrity of the LLM output (detecting that the generated text has been tampered). Public key detection of a fragile watermark has already been documented in the literature on watermarking (but not applied to LLM). Protecting the integrity is a valuable application, but this is very different from generated text detection.
> > >
> > > I can not accept a paper that sustains such confusion. The central claim of the paper (public-key detection of a robust watermark) is simply wrong.
> > >
> > > Regards

---

> > > > ### Author Response · Authors · 2024-09-25
> > > > **Further clarifications**
> > > >
> > > > To be completely clear: we do not claim that our watermark is robust in a general, arbitrary sense. We apologize if there is text in the paper that suggests otherwise—please let us know where and we will edit as necessary to avoid this interpretation. Such a strong notion of robustness has been clearly demonstrated to be impossible (Zhang et al., ICML 2024). In general, we would appreciate it if the reviewer has constructive and concrete suggestions.
> > > >
> > > > We do, however, claim that our watermark meets a very specific and weak notion of robustness. This notion is described exactly by Definition 3.7 and we prove that our scheme meets it in Theorem 4.8. Our goal was to demonstrate that public-detectability is possible in this model without introducing trivial forgery attacks. Moreover, our definitions are reflective of weak watermarks described by Zhang et al., which are the only form of watermark that can exist in light of their attacks. We took a bottom-up approach to text watermarking—rather than claiming that a watermark meets very strong, imprecise properties (which often turn out to be oversold, e.g., Kirchenbauer et al., 2023a, Kuditipudi et al., 2023, and Zhao et al., 2023a were all shown to be non-robust by Zhang et al.), we developed precise, weaker definitions and mathematically prove that our scheme meets them. We hope that this approach is a more productive way to incrementally and systematically build up to a practical watermark.
> > > >
> > > > Finally, even though there is much to be desired, we do believe that our scheme is useful for certain use-cases. One example would be as a method to prevent future models from training on watermarked text (Shumailov et al., 2023). Moreover, the scheme will improve as more techniques emerge. Edit distance robustness has already been demonstrated by Golovich-Moitra., 2024.

---

### Review · Reviewer_WcEH · 2024-07-16

**Summary Of Contributions:**

The paper presents a watermarking technique for Large Language Models (LLMs) with formal guarantees of soundness, completeness, and cryptographic distortion-freeness. Compared to previous work, such as Christ et al. (2023), this technique additionally ensures that the watermark is publicly detectable and that only the model provider can embed the watermark using rejection sampling.

**Audience:**

Yes

**Broader Impact Concerns:**

Broader impact statement is present

**Claims And Evidence:**

Yes

**Requested Changes:**

1. Further improve the writing of the paper for easier readability

**Strengths And Weaknesses:**

Strengths:
1. The paper proposes an LLM watermarking technique with formal guarantees of soundness and completeness
2. The approach provides guarantees on cryptographic distortion-freeness
3. The evaluation shows that the approach has small effect on the quality of LLM generated output and outperforms symmetric SOTA watermarking scheme from Christ et. al. 2023

Weakness:
1. The technique provides weak-robustness guarantee and points to the impossibility of strong robustness for watermarking LLM output. However, it would be good to see empirical robustness in the experiments.
2. The writing can be further improved, especially the section 2 on overview, with some examples to improve the readability of the paper. Especially for the TMLR readers from ML community who are not experts on cryptography.

Questions:

>“let σ := Signsk(H(t)).”

Don’t introduce key symbols in footnotes. If they are meant to be read, define them in the main text even if it is informal introduction.

>“Algorithm 1”

What’s n here? There are some symbols that are being reused, be careful in defining them

>“Algorithm 2,3, 4”

Can you explicitly define the inputs to algorithms again in the pseudocode. It becomes hard to keep track of all the symbols, and there needs to be too much back and forth to understand them.

>“Note that throughout the paper we have discussed embedding the signature in tokens for simplicity and alignment with prior work.”

Shouldn’t alignment with current work be more important than alignment with prior work? This change confuses me a bit, What are the consequences of embedding in characters instead of tokens?

>“We run asymmetric first, then use the number of tokens from the execution in the subsequent algorithms. This ensures that all algorithms produce the same number of tokens.”

In the evaluation, this step ensures that all the techniques generate same number of tokens, by forcing the model to not take the EOS token. I don’t understand how is this a fair evaluation for time comparison with other techniques or with plain generation.

---

> ### Author Response · Authors · 2024-08-01
> **On empirical robustness, characters vs. tokens, and time comparisons**
>
> **Empirical robustness.** We are happy to provide an empirical evaluation of robustness. Our weak robustness is described by Theorem 4.6: an adversary cannot change tokens that correspond to the message or signature component of the embedded gadget. We are confident that the evaluation will line up with this exactly.
>
> **Characters vs tokens.** The choice of implementing the algorithm at a token or character level makes no significant difference. The reason we implemented it at the character level was to simplify support for arbitrary tokenizers, which in turn allowed us to support any arbitrary Hugging Face-hosted language model. To implement the watermark, it is convenient if the tokenizer always parses the same string into the same tokens (i.e., it is deterministic). This is not true for most tokenizers: there are usually multiple valid tokenizations of the same string. Applying the watermark at the character level avoids this issue with no degradation.
>
> **Time comparison.** Essentially, for each of the decoding algorithms in Figure 3 we time how long it takes to generate a “same length” completion. Since all of the decoding algorithms are computationally distortion-free, this ends up just being a measure of how many tokens per second the decoding algorithm can support.

---

### Review · Reviewer_mhHs · 2024-07-20

**Summary Of Contributions:**

This paper proposes a text watermarking scheme that is publicly detectable in contrsst to much of prior work on text watermarking. The proposed scheme forms a message and signature using a private key, and optionally and error correcting encoding scheme from the first few tokens generated from the LM. This signature is embedded into the generated text that follows the text used for signature creation. Specifically, $\beta$ bits are encoded into subsequent $l$ token strings by retrying generating a candidate string until a hash collision is found. The error correction mechanism tries to ameliorate issues caused by low entropy continuations. The work identifies many desiderata for a watermark namely: cryptographic detectability, robustness, distortion freeness, unforgeability, model agnosticity, and public verifiability. It then provides technical arguments for how the proposed approach satisfies most of these desiderata by construction. The empirical analysis focuses mainly on distortion freeness and runtime and is compared against another similar private watermarking scheme from prior work. In the paper, "distortion free-ness" refers to the property that the generations with and without the watermarking scheme are not noticeably different. This property is measured by using GPT-4 as a judge and comparing the goodness scores against non-watermarked and other watermarking approaches.

**Audience:**

Yes

**Claims And Evidence:**

No

**Requested Changes:**

Please address the concerns above. Especially, improving the readability of the paper is important. Connecting figures, algorithms, and text cleanly would improve the paper quality.

I also think that more thorough empirical evaluation around detectability and distortion freeness will improve the paper.

**Strengths And Weaknesses:**

Strengths:

-- The paper is well motivated and the established desiderata are reasonable.

-- The algorithm provided is supported by strong technical arguments.

-- The algorithm is easy to implement and seems to satisfy many of the desiderata identified.

Weaknesses:

-- The approach seems to require generating long sequences to embed the signature.

-- I am not convinced by the definition of "distortion free-ness". By construction, most of the watermarking algorithms, do change the autoregressive token distributions at each step, so they have the potential to distort the sequences one would get without watermarking. This algorithm too modifies the autoregressive distribution in order to agree with the signatures. Signatures being random does ameliorate this concern but I think it is not sufficient to establish distortion free-ness. The empirical evaluation is not very convincing either. Instead of quality estimation via GPT4, it would be better to generate multiple samples from non-watermarked and watermarked approaches for a prompt and characterize similarity/overlap between the texts.

-- Similarly, a more thorough discussion on how detectablity is ensured would improve the paper. I am unable to convince myself about the absence of false positives. This directly connects to the entropy of autoregressive LM token distributions and the algorithms requiring long generations.

-- The writing is confusing at places. For example, the variable $t$ on line 3 in Alg. 5 does not seem to be used at all later in the algorithm. I don;t understand why $\gamma$ is being incremented in line 13 of Alg 4. Their is room for improvement in terms of readablity.

-- Algorithm 5 is extremely difficult to parse. To the extent that I understand vaguely form the text, how its supposed to work but can;t convince myself of its correctness by reading the algorithm.

---

> ### Author Response · Authors · 2024-08-01
> **On distortion-freeness, characterizing similarity, and detectability**
>
> **On distortion-freeness.** At a high level, one can understand the distortion-freeness of our scheme as follows. The generation of the next token can be seen as a *deterministic* process that (also) takes the randomness as input. Our watermarking scheme does not change this deterministic process but only changes how the randomness is sampled. Instead of sampling truly random strings, we sample (cryptographic) pseudo-random strings. The indistinguishability (i.e., distortion-freeness) between watermarked and unwatermarked models relies solely on the fact that true randomness is indistinguishable from pseudo-randomness.
>
> **Characterizing similarity/overlap of watermarked and non-watermarked text.** While it would certainly be possible to measure similarity/overlap in the suggested manner, we believe this does not capture the property that distortion-freeness is designed to provide. The text quality evaluation we performed can be summarized as follows: we generate both watermarked and non-watermarked text completions (all of which are formally distortion-free) from the same prompt and ask an LLM oracle (i.e., GPT-4) to rank them. This is similar to "arena style" ELO rankings of LLM text generations, except human comparisons are replaced with an LLM proxy. If distortion-freeness is in effect and the LLM oracle is unbiased, the scores should be distributed in the same way (uniformly at random). While it would be possible to reuse the same evaluation framework and additionally calculate generation similarities, we fear this is not an accurate measure of the watermarks’ main goal. Both our watermarking protocol and the Christ et al. protocol are designed to produce text that is sampled from the “same” distribution as the base model: this is a different goal from aiming to produce text that is as similar as possible to the base model.
>
> **Detectability/absence of false positives.** We would be happy to elaborate on this. We believe that strong detection guarantees are one of the main strengths that cryptography can bring to GenAI watermarking. Essentially, our scheme will see false positives at the same rate as the underlying signature scheme: if it was possible to find a forged watermark produced by our scheme, it would be possible to extract a forgery for the underlying signature scheme from the watermarked text. Thus, the false positive rate is as low as the possibility of forging a signature, which is guaranteed to be negligibly low. We will clarify this issue in the paper and we can add detectability measurements if desired: our raw evaluation data already contains this information.

---

> > ### Comment · Reviewer_mhHs · 2024-09-03
> > **Thank you for your response**
> >
> > Hello, thanks for your response and especially the expansion on detectability. That said, my overall impression about strengths and weaknesses of the paper in my review remains unchanged.

---

> > > ### Author Response · Authors · 2024-09-04
> > > **We are ready ton answer further questions**
> > >
> > > Thanks. We are here to answer further questions if you have.

---

### Author Response · Authors · 2024-08-01
**Overarching comments**

We thank all reviewers for their valuable feedback. We acknowledge recurring comments about readability here and address reviewer-specific comments below.

**General writing improvements.** We greatly appreciate the presentation feedback, especially given the interdisciplinary nature of the paper. We will (a) improve algorithm presentation to lessen the variable burden, (b) better connect figures with corresponding text, particularly in technical exposition such as Section 2, and (c) fix other presentation issues such as key symbols in footnotes and minor typographic errors.

References:

[1] Hanlin Zhang, Benjamin L Edelman, Danilo Francati, Daniele Venturi, Giuseppe Ateniese, Boaz Barak. Watermarks in the sand: Impossibility of strong watermarking for generative models. ICML 2024

[2] Miranda Christ, Sam Gunn, Or Zamir. Undetectable watermarks for language models. COLT 2024

[3] Chhavi Yadav, Amrita Roy Chowdhury, Dan Boneh, Kamalika Chaudhuri. FairProof: Confidential and Certifiable Fairness for Neural Networks. ICML 2024

[4] Noah Golowich and Ankur Moitra. Edit Distance Robust Watermarks for Language Models. arXiv preprint 2024

[5] Miranda Christ and Sam Gunn. Pseudorandom error-correcting codes. CRYPTO 2024

---

### Decision · Action_Editor_GPfk · 2024-09-24

**Recommendation:** Reject

**Comment:**

The reviewers shared several common concerns regarding the paper:

- Its current presentation and writing are not appropriate for the TMLR audience.
- There is insufficient numerical evidence to demonstrate that the proposed scheme effectively supports the claims made, particularly in terms of robustness and distortion-freeness. While the theoretical basis suggests this should be the case, it may only hold true for very long sequences.

Another reviewer also pointed out that a potential risk of confusion on the term robust.They argue that, in line with the literature on watermarking, the authors should describe their scheme as a fragile watermark rather than a robust one. I concur with the reviewer's opinion. In our opinion, the authors should acknowledge existing work on fragile watermarking and make the distinction clear.

To conclude, the paper brings an interesting contribution but substantial work is required before its potential acceptance to TMLR. Therefore, I recommend rejection but encourage the authors to revise their manuscript based in the light of the reviewers' feedback and resubmit their work.

**Audience:**

Some reviewers suggest that the paper is more suited to a venue focused on cryptography. While I agree with their opinion given the current state of the paper, I believe it could still be of interest to the machine learning community, especially considering the growing focus on LLMs and their protection.

**Claims And Evidence:**

The paper develops a new fragile watermark scheme for LLMs embedding a cryptographic signature detectable using a shared public key. The construction of this scheme is based on the assumption that token generation is associated with a distribution with a min-entropy lower bounded by $\alpha$. Based on this assumption, the authors show that their scheme satisfies important theoretical guarantees, i.e., it is distortion-free, correct, sound, and robust. Finally, they conduct numerical experiments aiming to illustrate the advantages of their approach.

**Resubmission Of Major Revision:**

The authors may consider submitting a major revision at a later time.